# The *Wnt/β-catenin/TCF/Sp5/Zic4* Gene Network That Regulates Head Organizer Activity in *Hydra* Is Differentially Regulated in Epidermis and Gastrodermis

**DOI:** 10.3390/biomedicines12061274

**Published:** 2024-06-08

**Authors:** Laura Iglesias Ollé, Chrystelle Perruchoud, Paul Gerald Layague Sanchez, Matthias Christian Vogg, Brigitte Galliot

**Affiliations:** Department of Genetics and Evolution, Institute of Genetics and Genomics (iGE3), Faculty of Sciences, University of Geneva, 30 Quai Ernest Ansermet, 1205 Geneva, Switzerlandchrystelle.perruchoud@unige.ch (C.P.); pglsanchez@gmail.com (P.G.L.S.);

**Keywords:** *Hydra* head organizer, *Hydra* transgenic lines, epidermal and gastrodermal epithelial layers, Sp5 transcription factor, *Sp5* promoter autoregulation, Zic4 transcription factor, Zic4 tentacle regulator, Wnt/β-catenin signaling, gene regulatory network (GRN), HEK293T mammalian cells

## Abstract

*Hydra* head formation depends on an organizing center in which Wnt/β-catenin signaling, that plays an inductive role, positively regulates *Sp5* and *Zic4*, with Sp5 limiting *Wnt3/β-catenin* expression and Zic4 triggering tentacle formation. Using transgenic lines in which the *HySp5* promoter drives eGFP expression in either the epidermis or gastrodermis, we show that *Sp5* promoter activity is differentially regulated in each epithelial layer. In intact animals, epidermal *HySp5:*GFP activity is strong apically and weak along the body column, while in the gastrodermis, it is maximal in the tentacle ring region and maintained at a high level along the upper body column. During apical regeneration, *HySp5*:GFP is activated early in the gastrodermis and later in the epidermis. Alsterpaullone treatment induces a shift in apical *HySp5:GFP* expression towards the body column where it forms transient circular figures in the epidermis. Upon *β-catenin*(RNAi), *HySp5:*GFP activity is down-regulated in the epidermis while bud-like structures expressing *HySp5:*GFP in the gastrodermis develop. *Sp5*(RNAi) reveals a negative *Sp5* autoregulation in the epidermis, but not in the gastrodermis. These differential regulations in the epidermis and gastrodermis highlight the distinct architectures of the *Wnt/β-catenin/TCF/Sp5/Zic4* network in the hypostome, tentacle base and body column of intact animals, as well as in the buds and apical and basal regenerating tips.

## 1. Introduction

*Hydra* is a freshwater hydrozoan polyp known for its exceptional regenerative capacities, including its capacity to regrow any missing part of its body, such as a new fully functional head in three to four days after a mid-gastric bisection (reviewed in [1]). Its anatomy is simple; it is a gastric tube composed of two myoepithelial layers known as the epidermis and gastrodermis along a single oral–aboral axis. This bilayered gastric tube connects the apical or head region at the oral side to the basal disc at the aboral side. The regenerative process relies on the rapid establishment of a head organizer in the regenerating tip, initially identified by Ethel Browne through transplantation experiments [2]. Indeed, she showed that tissues isolated from the head of intact animals, from the head-regenerating tip or from the presumptive head of the developing bud, can instruct and recruit cells from the body column of the host to induce the formation of an ectopic head, a property later named organizer activity by Spemann and Mangold [3]. Additional transplantation experiments confirmed that the head organizer is actively involved in developmental processes in *Hydra* such as 3D reconstruction of the missing head after decapitation at any level along the body column or formation of a new head during budding. In addition, the head organizer is also required in a homeostatic context, actively maintaining head patterning in intact animals [4,5,6,7,8,9]. Hence, in *Hydra*, two types of head organizer activity take place, one in homeostasis and the other in developmental contexts, the latter ones giving rise to the former.

The principle of organizer activity was later shown to be also at work during embryonic development in vertebrates, initially in gastrulae [3,10] and later on during appendage and hindbrain development [11,12,13,14]. These organizers are transient developmental structures with evolutionarily conserved patterning properties [15]. Similarly, regenerating blastema that form after amputation can be considered as organizing centers, which exhibit patterning properties to reconstruct the missing structure due to the molecular instructions they deliver to the surrounding cells to modify their behavior [8,16,17,18]. Indeed, these recruitment and patterning properties can be observed by transplanting regenerative blastema.

The *Hydra* polyp, an animal easily maintained in the laboratory, provides a model to decipher the cellular and molecular basis of regeneration. Transplantation experiments identified two distinct activities for the head organizer named head activation and head inhibition, both with maximal activity apically and a theoretical parallel apical-to-basal-graded distribution along the body axis [5,7,19,20]. In 1972, Gierer and Meinhardt proposed a reaction–diffusion model close to Turing’s one to predict how the organizer acts and how it is restored after bisection, with both processes relying on the cross-talk between an auto-catalytic short-range activator and a longer-range inhibitor, interacting in a positive–negative feedback loop [21]. According to this model, the activator positively acts on its own production as well as on that of the inhibitor, whereas the inhibitor negatively acts on the production/activity/stability of the activator. At any position along the animal length, the equilibrium between these two components is tightly controlled under homeostatic conditions, but is immediately disrupted upon amputation, resulting in the rapid restoration of the activity of the activating component, i.e., head activation, and the delayed restoration of the activity of the inhibitory component, i.e., head inhibition, given their respective rates of diffusion, self-regulatory capacity and cross-regulation.

Three decades later, Wnt/β-catenin signaling was proposed to actually act as the head activator, required to initiate apical morphogenesis and maintain apical differentiation in *Hydra* [22,23,24,25,26,27]. More recently, the transcription factor Sp5, whose expression is regulated by Wnt/β-catenin signaling in many species including *Hydra* [28,29,30,31,32,33], was shown to restrict the activity of Wnt/β-catenin signaling, thus fulfilling the expected positive–negative feedback loop of the head inhibitor [33]. Indeed, a transient knock-down of *Sp5* suffices to induce a multiheaded phenotype characterized by ectopic head formation along the body column of intact animals and the regeneration or budding of animals with multiple heads [33]. As anticipated, after a mid-gastric bisection, *Sp5* and *Wnt3* are up-regulated in the apical-regenerating tips, within two to three hours for *Wnt3* and after eight hours for *Sp5*, and both of them remain expressed at high levels throughout the entire head regenerative process but not the foot one [33]. Also, the transcription factor Zic4, whose gene expression is positively-regulated by Sp5, is responsible for the maintenance of tentacle differentiation and for their formation during apical development [34].

*Hydra* is populated by a dozen distinct cell types that are derived from three populations of adult stem cells, i.e., epithelial–epidermal, epithelial–gastrodermal and interstitial, which constantly self-renew in the body column to maintain *Hydra* homeostasis. In intact animals, *Wnt3*, *Sp5* and *Zic4* are predominantly expressed in epithelial cells of both the gastrodermis and epidermis [25,33,34], a finding confirmed by single-cell sequencing [35] (Appendix A). However, while *Wnt3*, *Sp5* and *Zic4* are expressed at their highest levels apically, their respective profiles in the apical region are very different: *Wnt3* is detected at a maximum level at the tip of the head, around the mouth opening, where the organizing activity is located [25,27]. In this region, *Sp5* and *Zic4* are in fact not detected, their expression being maximal at the base of the head where the tentacles are implanted and in the proximal region of the tentacles (Figure 1A). Also, *Sp5* is the only one to be detected along the body column.

Pharmacological and genetic manipulations have shown that dynamic interactions between Wnt/β-catenin, Sp5 and Zic4 play a crucial role for apical development and the maintenance of apical patterning [33,34]. However, although *Wnt3* is predominantly expressed in the gastrodermis, the specific role of each epithelial layer in the formation and maintenance of the head organizer remains unknown. The aim of this study is to uncover the dynamics of the Wnt/β-catenin/Sp5/Zic4 gene regulatory network in the epidermis and gastrodermis. To test these regulations in homeostatic and developmental contexts, we generated two transgenic lines that constitutively express the *HyAct-1388*:mCherry_*HySp5-3169*:GFP reporter construct in either epithelial layer. We monitored mCherry and GFP fluorescence in parallel with the detection of *GFP*, *Sp5*, *Wnt3* expression in intact, budding or regenerating animals, as well as in animals where Wnt/β-catenin signaling is either stimulated or knocked-down. In each of these contexts, we recorded distinct regulations of *Sp5* in the epidermis and gastrodermis, notably an epidermal-specific negative autoregulation of *Sp5* in the body column. These results point to distinct architectures of the *Wnt/β-catenin/Sp5/Zic4* gene regulatory networks active in the epidermis and the gastrodermis, and we discuss their respective roles in the regulation and activity of the head organizer.

## 2. Materials and Methods

### 2.1. Animal Culture and Drug Treatment

Strains of *Hydra vulgaris (Hv)* identified as *Basel (Hv_Basel)*, *magnipapillata* (*Hm-105*) or *AEP2 (Hv_AEP2)* strains [36] were cultured in Hydra Medium (HM: 1 mM NaCl, 1 mM CaCl2, 0.1 mM KCl, 0.1 mM MgSO4, 1 mM Tris pH 7.6) at 18 °C and fed two to three times a week with freshly hatched *Artemia* nauplii (Sanders, Aqua Schwarz). For regeneration experiments, animals were starved for four days, then bisected at mid-gastric bisection. To activate Wnt/β-catenin signaling, *Hv_Basel* or *Hv_AEP2* animals, starved for three or four days, were treated with 5 µM Alsterpaullone (ALP, Sigma-Aldrich A4847, St. Louis, MO, USA) or 0.015% DMSO for the indicated periods of time and subsequently washed with HM.

### 2.2. Mapping of the Transcriptional Start Sites (TSS)

*Sp5* and *Zic4* cDNAs sequences obtained by high throughput sequencing, available on HydrAtlas [37], Uniprot or NCBI, were aligned to the corresponding *Hm-105* genomic sequences (Appendix A) with the Muscle Align program (ebi.ac.uk/Tools/msa/muscle/) selecting a ClustalW output format. Next, the alignment was visualized with the MView tool [38] (ebi.ac.uk/Tools/msa/mview/) and the putative TSS deduced from the 5′ end of cDNAs.

### 2.3. Reporter Constructs Expressed in Hydra or in HEK293T Cells

All reporter constructs used in this study are listed in Appendix A. To produce the *HyAct-1388:*mCherry_*HySp5-3169*:eGFP construct (further named *HySp5-3169:GFP)*, a block of 3194 bp *HySp5* sequences were amplified from *Hm105* genomic DNA including 2968 bp promoter sequences, 201 bp 5′UTR sequences and 27 bp coding sequences. The sequences of the *HySp5* promoter Forward and Reverse primers are given in Appendix A. The hoTG-*HyWnt3FL*-EGFP-*HyAct*:dsRED plasmid (kind gift from T. Holstein) [27] was then digested with the EcoRV and AgeI enzymes to remove the *HyWnt3FL* promoter region and insert the *Sp5* 3194 bp region. Next, the dsRED sequence was replaced by the mCherry sequence by GenScript. The sequences of the final construct were verified by sequencing (Appendix A). From the *HySp5-2992*:Luciferase construct [33], six constructs were generated either by deleting the five Sp5-binding sites (BS) located in the proximal promoter or by mutating one of them: BS1 at position −129, BS2 at position −105, BS3 at position −52, BS4 at position −34 and BS5 at position +18. These *HySp5-2828*:Luc, *HySp5-2992-mBS1*:Luc, *HySp5-2992-mBS2*:Luc, *HySp5-2992-mBS3*:Luc, *HySp5-2992-mBS4*:Luc and *HySp5-2992-mBS5*:Luc constructs were generated using the QuikChange Lightning Multi Site-Directed Mutagenesis Kit (Agilent Technologies, Santa Clara, CA, USA).

### 2.4. Generation of the Hydra Transgenic Lines

To generate the *Sp5* transgenic lines, gametogenesis was induced in the *Hv_AEP2* strain by alternating the feeding rhythm from four times per week to once a week consequently. The *HySp5-3169*:GFP construct was injected into one- or two-cell-stage *Hv_AEP2* embryos [39]. Out of 330 injected eggs, 27 embryos hatched and 3/27 embryos exhibited GFP and mCherry fluorescence. The epidermal and gastrodermal *HySp5*-3169:GFP lines analyzed in this work were obtained through clonal propagation from a single embryo for each of them in which only a few cells were positive after hatching. By asexual reproduction of the original animal, i.e., budding, we obtained two transgenic animals with a complete set of mCherry–eGFP-positive epithelial cells, either epidermal or gastrodermal. The generation of the epidermal and gastrodermal *HyWnt3FL*:eGFP*-HyAct*:dsRED transgenic lines, renamed here as epidermal *HyWnt3*-2149:GFP and gastrodermal *HyWnt3*-2149:GFP, is described in [33].

### 2.5. RNA Interference

For the gene silencing experiments, we applied the procedure reported in [33]. Briefly, four-day starved budless animals were selected from the *Hv_AEP2* culture, rinsed 3× in water, incubated for 45–60 min in Milli-Q water and electroporated with 4 µM siRNAs, either targeting *Sp5* or *β-catenin* or scramble as the negative control. For *Sp5* and *β-catenin,* an equimolecular mixture of three siRNAs was used (siRNA1+siRNA2+siRNA3, see sequences in Appendix A). Animals were electroporated once, twice or three times (EP1, EP2 and EP3) every other day as indicated.

### 2.6. Quantitative RT-PCR

At the indicated time-points after electroporation, 20 animals per condition were amputated either at an 80% level to obtain the apical region (100–80%) and the body column (80–0%) or at 80% and 30% levels to obtain the apical region as above, the central body column (80–30%) and the basal region (30–0%). Besides electroporated transgenic animals, non-electroporated wild-type *Hv_AEP2* animals were used to provide the reference expression levels. The different parts of the animals were transferred to RNA-later (Sigma-Aldrich R0901) immediately after amputation and kept at 4 °C prior to RNA extraction. RNA extraction was performed using the E.Z.N.A.^®^ Total RNA kit (Omega, Norcross, GA, USA) and cDNA was synthesized with the qScript^TM^ cDNA SuperMix (Quanta Biosciences, Beverly Hills, CA, USA). The cDNA samples were diluted to 1.6 ng/µL and the primer sequences used to amplify the *Sp5*, *Wnt3*, *β-catenin*, *GFP* and *TBP* genes were designed with Primer3-OligoPerfect (Thermo Fisher, Waltham, MA, USA) (Appendix A). Quantitative RT-PCR was performed using the SYBR Select Master Mix for CFX (Applied Biosystems, Waltham, MA, USA) and a Biorad CFX96^TM^ Real-Time System. Relative gene expression levels were calculated as described in [40], using *TBP* to normalize all data. Fold change (FC) values at each time point or condition were calculated over the values obtained in non-electroporated animals. Finally, in each condition, the FC values measured in *β-catenin*(RNAi) or *Sp5*(RNAi) animals were divided by those measured in animals of the same condition exposed to scramble dsRNA.

### 2.7. Whole-Mount In Situ Hybridization (WM-ISH)

The animals were relaxed in 2% urethane/HM for 1 min, fixed in 4% PFA prepared in HM for 4 h at room temperature (RT), then washed several times with MeOH before being stored in MeOH at −20 °C. WMISH was performed as described in [33]. For double WMISH, the *Wnt3* riboprobe was labeled with DIG (Sigma, Roche-11277073910) and the *Sp5* and *GFP* riboprobes were labeled with fluorescein (Sigma, Roche-11685619910); the *Wnt3*-DIG riboprobe was co-incubated with either the *Sp5*-FLUO or *GFP*-FLUO riboprobe during the hybridization step. The *Wnt3*-DIG riboprobe was first detected with NBT/BCIP (Sigma, Roche-11383213001) and the FLUO-labeled riboprobe was subsequently detected with Fast Red. To stop the NBT/BCIP reaction, the samples were washed several times in NTMT, then incubated in 100 mM glycine, 0.1%Tween (pH 2.2) for 10 min, washed in Buffer I (1× MAB; 0.1% Tween), then incubated in Buffer I supplemented with 10% sheep serum (Buffer I-SS) for 30 min at RT and prolonged for 1 h with fresh Buffer I-SS at 4 °C. Incubation with the anti-FLUO-AP antibody (1:4000, Roche-1142638910) was carried out at 4 °C overnight. Next, the samples were briefly washed in Buffer I then in 0.1M Tris/HCl (pH 8.2) 3× 10 min and then developed with Fast Red (SigmaFAST, F4648, St. Louis, MO, USA). To stop the reaction, the samples were washed several times in 0.1M Tris/HCl (pH 8.2) and fixed in 3.7% formaldehyde for 10 min at RT, rinsed in water and mounted in Mowiol. The co-detection of two riboprobes is technically challenging as the NBT/BCIP detection of the DIG-labeled riboprobe, which is normally far more sensitive than the Fast Red detection of the fluorescein-labeled riboprobe, is much less efficient when tissues are treated for Fast Red. Consequently, we first analyzed the expression pattern of each gene separately and subsequently co-detected *Wnt3* and *Sp5*, or *Wnt3* and *GFP*. In these conditions, we found the co-detection highly informative to record context-specific regulations. The plasmids used to produce the riboprobes are listed in Appendix A.

### 2.8. Immunofluorescence

Animals were fixed and subsequently rehydrated as for WMISH with repeated washes in successive dilutions of EtOH in PBST (PBS, 0.5% Triton). Blocking was performed with 2% BSA in PBST for 1–2 h at RT. For immunostaining, samples were incubated overnight at 4 °C with an anti-GFP antibody (1:400, Novus NB600-308) in 2% BSA. Then, after several washes in PBST, the secondary antibody anti-rabbit coupled to Alexa 488 (1:600, Invitrogen A21206) was added in 2% BSA for 4 h. For double immunofluorescence, the anti-mCherry (1:400, Abcam ab125096) and secondary anti-mouse coupled to Alexa 555 (1:600, Invitrogen A31570) antibodies were used.

### 2.9. Nuclear Extracts (NEs) and Electro-Mobility Shift Assay (EMSA)

NEs were prepared according to [41]. Briefly, 100 *Hm-105 or Hv_AEP2* animals were washed rapidly in HM and once in Hypotonic Buffer (HB: 10 mM Hepes pH7.9, 2 mM MgCl_2_, 5 mM KCl, 0.5 mM spermidine, 0.15 mM spermine), then placed in a 1 mL glass dounce with 1 mL HB and 20 strokes were given. After slowly adding (drop by drop) 210 µL of 2 M sucrose, 15 more strokes were given. The extract was centrifuged for 10 min at 3200 rpm at 4 °C, the pellet was washed twice with 800 µL Sucrose Buffer (0.3 M sucrose in HB), resuspended in 50 µL Elution Buffer (glycerol 10%, 400 mM NaCl, 10 mM Hepes pH 7.9, 0.1 mM EDTA, 0.1 mM EGTA, 0.5 mM spermidine, 0.15 mM spermine) and incubated for 45 min. The eluate was centrifuged at 4 °C for 20 min at 13,000 rpm and the supernatant was aliquoted and stored at −80 °C. All manipulations were carried out on ice and all buffers contained a mix of a protease inhibitor cocktail (Bimake B14012).

The LightShift Chemiluminescent EMSA Kit (Thermo-Scientific, 89880, Waltham, MA, USA) was used to perform EMSA with *Hydra* NE and biotin-labeled double-stranded oligos (Appendix A). Briefly, 3 µL of *Hydra* NEs per 20 µL binding reaction was incubated for 20 min with the double-stranded oligos (20 fmol), then loaded on a 6% polyacrylamide gel (PAGE) in TBE 0.5x, electrophoresed and transferred onto a nylon membrane (BrightStar-Plus Invitrogen AM10102). A crosslink was carried out by exposing the membrane to UV-light 120 mJ/cm^2^ for 30–50 s (Marshall Scientific, SS-UV1800, Hampton, NH, USA) and the samples fixed on the membrane were blocked for 15 min with Blocking Buffer (Thermo-Scientific 89880A). The membrane was conjugated with Stabilized Streptavidin-Horseradish Peroxidase (1:300, Thermo-Scientific, 89880D) and developed by adding a Luminol/Enhancer solution (Thermo-Scientific 89880E/F).

### 2.10. Production of Anti-Sp5 Antibodies

Two anti-Sp5 antibodies were generated. A rabbit polyclonal antibody was produced by Covalab (Bron, France) against three peptides: P1 (178–191), NEHHIKEYSEHSQA; P2 (398–411), CDENVMELEVNVEN; and P3 (155–175), PASPISWLFPQNIIQSHPSKV. After four immunizations, the sera were collected from a single rabbit and an ELISA test was performed to check the immunoreactivity. Next, the sera were purified by Covalab with the peptides P1 and P2 to remove any P3 cross-reactivity. The mouse monoclonal antibody was produced by Proteogenix (Schiltigheim, France) against a 6His-tag (MGSHHHHHHSG) coupled to a 218 AA-long *Hydra* Sp5 fragment (ISPLEQT---YSMSTSI) produced chemically. The Sp5-218 protein (24.5 kDa) was expressed in E. coli and injected into the animals. After four immunizations, spleen cells collected from two mice were fused to myeloma cells. The antibody, produced from one selected clone, was validated by IP analysis.

### 2.11. Cell Culture and Whole Cell Extracts (WCEs) and Western Blotting

The immortalized human embryonic kidney HEK293T cells were cultured in DMEM High Glucose, 10% fetal bovine serum (FBS), 6 mM L-glutamine and 1 mM NA pyruvate in 10 cm-diameter cell culture dishes (CellStar, Greiner Bio-One 664160, Kremsmünster, Austria). After a two-day growth, the cells were collected by scraping, counted and 15 × 10^4^ cells per well were seeded in 6-well plates and grown for 19 h. Next, the cells were transfected with 2 µg of pCS2+empty or pCS2+*HySp5* plasmid using the X-tremeGENE HP DNA transfection reagent (Sigma, 6366546001, St. Louis, MO, USA). To prepare the cell extracts 24 h later, the cells were resuspended in PBS 1× before being centrifuged for 3 min at 3000 rpm at 4 °C. After discarding the supernatant, the pellet was resuspended in fresh Lysis Buffer (LB): 50 mM Hepes pH 7.6, 150 mM NaCl, 2.5 mM MgCl_2_, 0.5 mM DTT, 10% glycerol, 1% Triton 100×, 0.1 mg/mL PMSF, 10% protease inhibitor cocktail (Bimake, B14012, Houston, TX, USA) and a lab-made phosphatase inhibitors cocktail (8 mM NaF, 20 mM β-glycerophosphate, 10 mM Na_3_VO_4_). After a 30 min incubation on ice, the extract was centrifuged at 14,000 rpm for 10 min at 4 °C and the supernatant was aliquoted and stored at −80 °C. Next, 20 µg extracts, either WCEs or NEs, were diluted with Loading Laemmli buffer and then boiled for 5 min at 95 °C before being loaded onto a 10% SDS-PAGE, then electrophoresed and transferred onto a PVDF membrane (Bio-Rad 162-0177). Next, the membrane was blocked for 1 h at RT with 5% dry milk in TBS 1×, 0.1%Tween (TBS-T). Anti-Sp5 antibodies were added at a 1:500 dilution and incubated overnight at 4 °C. The membranes were washed 3× 10 min in TBS-T before being incubated for 2 h with the secondary anti-mouse-HRP- or anti-rabbit HRP antibody (1:5000, Promega anti-mouse, W4021; anti-rabbit W4011). The membranes were washed in TBS-T for 3× 10 min and developed with Western Lightning Plus-ECL reagent (Perkin Elmer NEL104). To produce the Sp5 protein in vitro, the pCS2+empty and pCS2*HySp5 plasmids were incubated using the TNT Quick Coupled Transcription/Translation Systems (Promega L2080, Madison, WI, USA) and 1 µL was loaded on 10% SDS-PAGE.

### 2.12. Chromatin Immuno-Precipitation and Quantitative PCR (ChIP-qPCR)

ChIP was performed with 300 *Hm105 or Hv_AEP2* animals fixed in 1% Formaldehyde Solution (Thermo-Scientific 28906) for 15 min, then transferred into a Stop Solution (Active Motif 103922) for 3 min, briefly washed in cold HM before being resuspended in 5 mL Chromatin prep buffer containing 0.1 mM PMSF and 0.1% protease inhibitor cocktail (Active Motif 103923). The samples were transferred to pre-cooled 15 mL glass dounces and crushed with 30 strokes. The samples were incubated on ice for 10 min before being centrifuged at 4 °C for 5 min at 1250 rcf. Each pellet was resuspended in 1 mL Sonication Buffer (SB: 1% SDS, 50 mM Tris-HCl pH 8.0, 10 mM EDTA pH 8.0, 1 mM PMSF, 1% protease inhibitor cocktail) and incubated on ice for 10 min. The chromatin was then sonicated with a Diagenode Bioruptor Cooler (sonication conditions: Amp: 25%, Time: 20 s on, 30 s off, 2 cycles). The samples were centrifuged at 14,000 rpm for 10 min at 4 °C, the supernatant was sonicated (sonication conditions as above, but 3 cycles), centrifuged at 14,000 rpm for 10 min at 4 °C and the supernatant was recovered. After measuring the DNA with Qubit, 10 µg of the sonicated chromatin was diluted (1:5) in ChIP Dilution Buffer (DB: 0.1% NP-40, 0.02 M Hepes pH 7.3, 1 mM EDTA pH 8.0, 0.15 M NaCl, 1 mM PMSF, 1% protease inhibitor cocktail) and incubated with 1 µg of either the monoclonal or polyclonal α-Sp5 antibody or pre-immune serum antibody overnight at 4 °C on a rotating wheel. The sample was then loaded onto a ChIP-IT ProteinG Agarose Column (Active Motif 53039, Carlsbad, CA, USA), incubated on a rotating wheel for 3 h at 4 °C and washed 6 times with 1 mL Buffer AM1 before being eluted with 180 µL Buffer AM4. After, 1M NaCl and 3× TE buffer were added to perform decrosslinking overnight at 65 °C. Next, RNAse A (10 µg/µL) was added for 30 min at 37 °C followed by Proteinase K (10 µg/µL) for 2 h at 55 °C. Finally, the MiniElute PCR purification kit (Qiagen, 28004, Hilden, Germany) was used to purify the samples. DNA was eluted in 30 µL and 1 µL per condition that was used for qPCR.

### 2.13. Imaging

Live imaging to analyze the dynamics of mCherry and GFP fluorescence and the imaging of immunofluorescence on whole animals were performed on the Leica DM5500 microscope (Wetzlar, Germany). To quantify GFP fluorescence, the acquired data were analyzed with the Fiji ImageJ2 software. Optical sections were acquired using a Spinning disc confocal CSU (Yokogawa, Japan) mounted on an inverted Nikon Ti microscope (Tokyo, Japan) with both the bright-field and GFP channels merged. A confocal LSM780 Zeiss microscope (Oberkochen, Germany) was used to image the immunostained hypostome region of transgenic animals, as well as the budding region of live transgenic animals incubated in 1 mM linalool in HM for 10 min prior to imaging, then kept in the linalool solution between two coverslips separated by a 0.025 mm spacer. WMISH pictures were acquired with the Olympus SZX10 microscope.

### 2.14. Statistical Analyses

The statistical analyses were two-tailed, unpaired and were carried out using the GraphPad Prism 8.4.3 software. *p* values are for **** ≤ 0.0001, *** > 0.0001 and ≤0.001, ** > 0.001 and ≤0.01, * > 0.01 and <0.05 and ns ≥ 0.05.

## 3. Results

### 3.1. Differential Sp5 Regulation in the Epidermal and Gastrodermal Layers along the Body Axis

To monitor the regulation of *Sp5* expression in the epidermal and gastrodermal epithelial layers, we produced a tandem reporter construct, *HyAct-1388*:mCherry_*HySp5-3169*:GFP, where the *Hydra Actin* promoter (*HyAct*, 1388 bp) drives the ubiquitous expression of mCherry and the *Hydra Sp5* promoter (*HySp5*, 3169 bp) drives eGFP expression (Figure 1B and Appendix A). After injecting the reporter construct into *Hv_AEP2* embryos, two transgenic lines were obtained by clonal amplification, one expressing the reporter in the epidermis (epidermal *HySp5-3169*:GFP) and the other in the gastrodermis (gastrodermal *HySp5-3169*:GFP) (Figure 1C,D and Appendix A). Next, in the q-PCR analysis we compared the expression levels of *Sp5*, *GFP* and *Wnt3* in the apical, central body column and basal regions of each transgenic line (Figure 1E,F). As expected, in both lines, we found *Wnt3* expressed exclusively apically and *Sp5* expressed in all regions but at maximal levels apically. By contrast, *GFP* appears differentially regulated along the two layers of the body axis, rapidly declining in its expression from the apical region to the upper gastric column in epidermal *HySp5-3169*:GFP animals, with a similar high level of expression in the apical and upper body column regions in gastrodermal *HySp5-3169*:GFP animals, followed by a low level of expression in the basal region (Figure 1E,F). These results indicate that the *Sp5*-3169 promoter is differentially regulated along the body column in the epidermal and gastrodermal layers.

**Figure 1 biomedicines-12-01274-f001:**
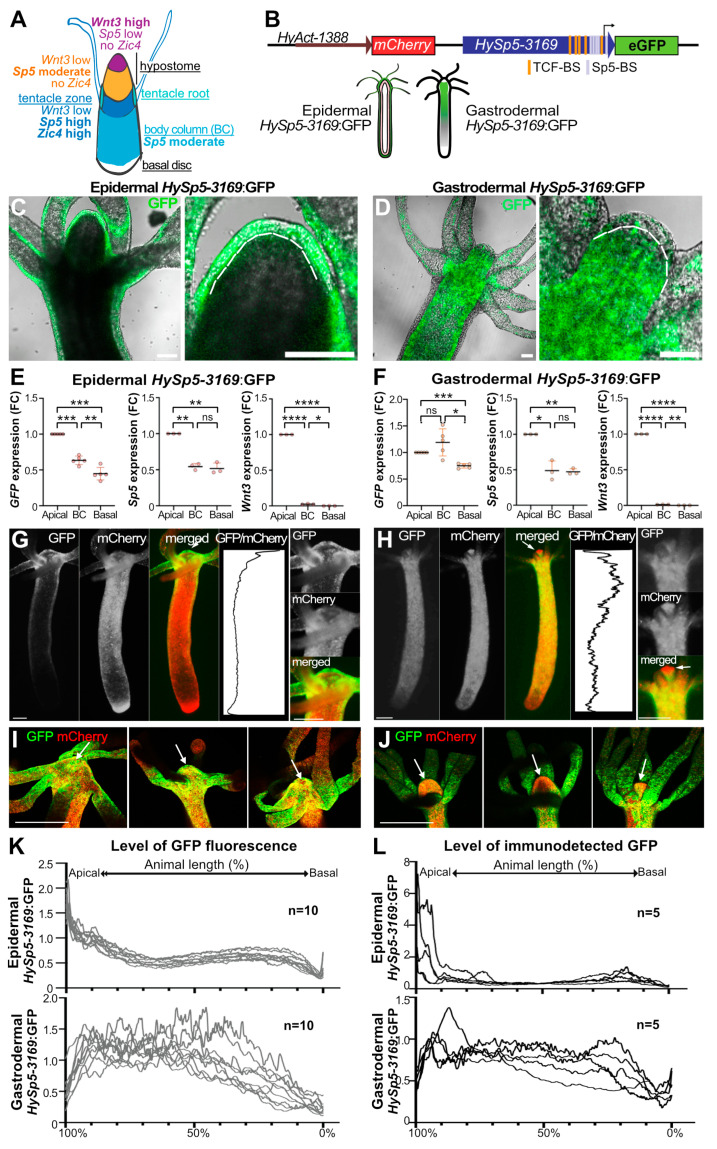
Differential regulation of *HySp5-3169*:GFP expression in the epidermal and gastrodermal layers. (**A**) Schematized view of *Hydra* anatomy that includes the apical region or head, formed from a dome-shape named hypostome, centered around the oral opening, surrounded by a ring of tentacles at its basis, the elongated or contracted body column and the basal disc or foot that can attach to substrates. The respective expression levels of *Wnt3*, *Sp5* and *Zic4* define three distinct domains in the apical region (see ref. [34]). (**B**) Structure of the *HyAct-1388*:mCherry_*HySp5-3169*:GFP reporter construct used to generate the epidermal and gastrodermal *HySp5-3169*:GFP transgenic lines, where epithelial cells from the epidermis and gastrodermis, respectively, express GFP and mCherry (sequence in Appendix A). TCF-BS: TCF-binding sites (orange); Sp5-BS: Sp5-binding sites (grey). (**C**,**D**) Optical sections of live transgenic animals expressing *HySp5*-3169:GFP in the epidermis (**C**) and gastrodermis (**D**). Brightfield and eGFP channels are shown and the apical region of each animal is magnified on the right (Appendix A). The dashed white line indicates the thin extracellular layer known as the mesoglea, which delimits the epidermal (outer) and gastrodermal (inner) epithelial layers. Scale bar: 100 µm. (**E**,**F**) Q-PCR analysis of *GFP*, *Sp5* and *Wnt3* transcript levels in the apical, body column (BC) and basal regions of epidermal (**E**) and gastrodermal (**F**) *HySp5-3169*:GFP animals fixed immediately after dissection. *p* values as indicated in Materials & Methods. (**G**,**H**) Live imaging of epidermal (**G**) and gastrodermal (**H**) *HySp5-3169*:GFP animals with eGFP (green), mCherry (red); the apical region of each animal is magnified on the right. White arrows point to the tip of the hypostome. The graphs show the eGFP/mCherry fluorescence intensity ratios (relative GFP intensity) along the animal axis. (**I**,**J**) Immunodetection of GFP (green) and mCherry (red) in the apical region (white arrow) of epidermal (**I**) and gastrodermal (**J**) *HySp5-3169*:GFP animals. Scale bars in panels (**G**–**J**): 250 µm. (**K**,**L**) Graphs showing the relative GFP fluorescence recorded live (**K**) or after immunodetection (**L**) in epidermal or gastrodermal *HySp5-3169*:GFP animals. The apical extremity is on the left (100%), the basal one on the right (0%). See Appendix A.

This result was confirmed at the protein level by recording live fluorescent GFP (Figure 1G,H and Appendix A) or by immunodetecting GFP (Figure 1I,J and Appendix A). In epidermal transgenic animals, GFP fluorescence and GFP protein are detected in the epidermal layer over the whole hypostome, the tentacle ring, the proximal part of the tentacles and the upper body column (Figure 1G,I and Appendix A). In gastrodermal transgenic animals, GFP fluorescence and GFP protein extend over a broad domain in the gastrodermal layer, from the apical region throughout the body column (Figure 1H,J and Appendix A). However, the tip of the hypostome is free of gastrodermal GFP fluorescence and GFP protein (see enlarged head in Figure 1H, arrows in Figure 1J), an area where *Wnt3* expression is maximal and endogenous *Sp5* expression is minimal [27,33].

Together with the colocalization of endogenous *Sp5* transcripts and immunodetected *HySp5-3169*:GFP (Appendix A), these analyses show that 3169 bp sequences of the *Sp5* promoter are sufficient to recapitulate the previously identified endogenous *Sp5* expression pattern in the apical region and along the body column [33] and to highlight previously unrecognized differences in the expression between the epidermis and gastrodermis.

From the GFP and mCherry fluorescence profiles, we produced a relative GFP intensity profile for each animal that corresponds to the GFP/mCherry ratio at any point along the body axis (Figure 1K,L and Appendix A). By superimposing the profiles of 10 animals, we concluded that GFP fluorescence in the epidermis of live animals is graded apical-to-basal, from 100% to 70% of the body length, then maintained at low levels between the 70% to 10% positions (Figure 1K and Appendix A). In contrast, in gastrodermal_*HySp5-3169*:GFP animals, the GFP levels are low at the most apical end (position 100–90%), reach a high plateau from the position of 90% up to 40% of the body length and then decrease towards the basal end (Figure 1K and Appendix A). For each transgenic line, we noted that the fluorescence intensity profiles of GFP in live animals corresponded to the profile of GFP immunodetected in the corresponding fixed samples (Figure 1L). In conclusion, a comparative analysis of *GFP* transcripts, GFP fluorescence and GFP protein converges to identify distinct patterns in the epidermis and gastrodermis, both apically and along the body axis, indicating that *Sp5* is differentially regulated in these two layers.

### 3.2. Sp5 Regulation after Bisection Is Systemic in Gastrodermis but Localized in Epidermis

Next, we analyzed how *HySp5-3169*:GFP is regulated in developmental contexts. During regeneration, epidermal *HySp5-3169*:GFP animals show no or low *GFP* expression in their apical-regenerating (AR) tips fixed at 8 and 12 h post-amputation (hpa) (Figure 2A and Appendix A). Then, at 24 hpa, *GFP* expression is detected. At 48 hpa, tentacle rudiments that emerge do not express *GFP* whereas the tip of the developing hypostome strongly expresses *GFP*; at 72 hpa, the epidermal *GFP* pattern is typical, with maximal expression at the root of tentacles. In contrast, in gastrodermal *HySp5-3169*:GFP animals, *GFP* is detected immediately after bisection, presumably artifactually in injured tissue, then at high levels at 8 and 12 hpa in a broad domain encompassing the AR tips (Figure 2B and Appendix A). At 24 hpa, the gastrodermal *GFP* expression becomes restricted to the apical area, at 48 hpa the emerging tentacles and the tip of the future hypostome are free of *GFP* expression. At 72 hpa, apical *GFP* expression is mainly present in the tentacle ring, absent from the tentacles and hypostome and at a low level in the peduncle region. Along the body column of these animals, the gastrodermal *GFP* expression either forms stripes alternating higher and lower levels or is diffused throughout the animal.

During basal regeneration, *GFP* expression is excluded from the basal-regenerating (BR) half in epidermal_*HySp5-3169*:GFP animals at any time-point (Figure 2A and Appendix A). At 48 hpa, most animals have differentiated a new basal disc and *GFP* expression is slightly up-regulated in the peduncle region. In gastrodermal *HySp5-3169*:GFP animals, the immediate *GFP* signals observed in the BR tips is presumably artifactual, linked to injury as in AR tips (Figure 2B and Appendix A). At subsequent stages, *GFP* expression is quite strong in the body column, in continuity with the apical domain, but becomes weaker at 24 hpa. In the BR tips, gastrodermal *GFP* expression is transient, becoming low or undetectable in most animals at 24 hpa. A new basal disc, free of *GFP*, is usually formed at 48 hpa, whereas some *GFP* expression remains in the adjacent peduncle region.

Regarding GFP fluorescence, it can be detected during apical regeneration at high levels in both epithelial layers at 8, 12 and 24 hpa, extending along the AR half, except in the peduncle region (Figure 2C,D and Appendix A). Subsequently, as the tentacle rudiments appear, epidermal GFP fluorescence becomes maximal in the apical region, while gastrodermal GFP fluorescence disappears from the tip of the forming head and becomes predominant in the tentacle ring and upper body column, resembling the homeostatic pattern. As expected, in epidermal *HySp5-3169*:GFP animals regenerating their basal half, no GFP fluorescence is observed, except in the apical region of origin (Figure 2C and FAppendix A); in gastrodermal *HySp5-3169*:GFP animals, GFP fluorescence is widely distributed along the body axis, but excluded from the differentiating basal disc, as observed at 48 hpa (Figure 2D and Appendix A).

During budding, GFP fluorescence is detected throughout the process in the epidermal and gastrodermal layers of *HySp5-3169*:GFP animals, but with different patterns (Figure 2E–H). In the budless parental polyp, epidermal GFP fluorescence is first visible as a patch preceding bud formation (stage 1, Figure 2E), then in the budding belt, where GFP fluorescence persists until stage 6, it is seen as gradually forming well-defined boundaries (Figure 2E,F). In the gastrodermis, GFP fluorescence is also detected in the budding belt, but with diffuse boundaries, in continuity on the apical side with the GFP expression domain in the body column (Figure 2G,H). In the growing bud, GFP fluorescence is ubiquitously expressed in both layers, predominantly apical in the epidermis from stage 6 onwards. At stages 9 and 10, when the bud is mature and ready to detach, the epidermal and gastrodermal GFP fluorescence correspond to those observed in adult polyps, apically restricted in the epidermis, while they diffuse along the axis in the gastrodermis (Figure 2E,G).

In conclusion, the *HySp5-3169*:GFP transgenic lines highlight the temporal and spatial layer-specific regulations of *Sp5* linked to regeneration and budding. During apical regeneration, *Sp5* is up-regulated in the epidermis at an early–late phase (24 hpa) and not at all during basal regeneration. In contrast, gastrodermal GFP expression is broadly enhanced at an early stage, whatever the type of regeneration, apical or basal. This widespread increase in *GFP* expression in the gastrodermis reflects a systemic *Sp5* response to amputation, specifically driven by the *Sp5* promoter as such an increase is not observed with *mCherry* driven by the *Actin* promoter. Similarly, during budding, *Sp5* expression is tightly regulated in the epidermis, but rather diffuse and systemic in the gastrodermis.

### 3.3. Layer-Specific Modulations of Sp5 Expression upon Alsterpaullone (ALP) Treatment

We then compared the phenotypic changes induced by the GSK3β inhibitor Alsterpaullone (ALP), which in the *H. vulgaris* Zürich L2 strain (*Hv_ZüL2*) leads to an increase in the level of nuclear β-catenin in the body column and the subsequent activation of Wnt3/β-catenin signaling [23] (Figure 3A). As a result, a two-day ALP treatment induces the formation of multiple ectopic tentacles along the body column of *Hv_ZüL2* or *Hv_Basel* animals [23,33]. However, in animals from the *Hv_AEP2* strain, two-, four- or even seven-day ALP treatment only leads to the transient and partial development of a few ectopic tentacles along the body column, likely as a result of the lower sensitivity of *Hv_AEP2* animals to drug treatments [42].

Nevertheless, after a four-day treatment, we noticed additional morphogenetic changes such as the striking reduction in the size of both the original tentacles and the hypostome at the apical pole, together with the enlargement of the upper body column that appears globally “swollen”, the progressive disappearance of the basal disc and the refinement of the basal extremity (Figure 3B and Appendix A). Given the positive feedback loop that operates between *Wnt3/β-catenin*, *Sp5* and *Zic4* and the negative one between Sp5 and *Wnt3*, we investigated how *Sp5* expression is modulated in each epithelial layer when β-catenin signaling is constitutively activated. We thus exposed non-transgenic and *HySp5-3169*:GFP transgenic animals to ALP for two or four days and analyzed the concomitant modulations of *Sp5* and *Wnt3* and *GFP* and *Wnt3* (Figure 3B and Appendix A).

After two days, we observed in all conditions a transient extension of the apical *Sp5* and *HySp5-3169*:*GFP* expression domain (i.e., positive for both *Sp5* and *GFP*) below the head, forming a second tentacle ring, from which, in some animals, ectopic tentacles transiently emerge. In the body column, *Sp5* is globally up-regulated along the gastrodermis while about half of the epidermal *HySp5-3169*:*GFP* animals form circular figures along the upper body column, possibly outlining regions where ectopic structures are transiently induced (Figure 3B and Appendix A). We also noted, in both epidermal and gastrodermal *HySp5-3169*:*GFP* animals, a high level of *GFP* expression close to the basal extremity, including a *GFP+* ring just above the basal disc.

After four days, most of the two-day ALP-induced changes had vanished: *Sp5* and *GFP* were no longer detected apically, neither in the epidermis, nor in the gastrodermis, the *Sp5*/*GFP* epidermal figures along the body column had disappeared and the global epidermal *GFP* expression was dramatically reduced. In the gastrodermis, the *HySp5-3169*:*GFP* expression remained present in the central part of the body column in most animals (Figure 3B and Appendix A). In summary, this analysis shows a similar silencing of epidermal and gastrodermal *HySp5-3169:GFP* in the apical region, but striking differences along the body column, with *HySp5-3169:GFP* transiently enhanced and forming circular figures in the epidermis, while remaining diffuse and long-lasting in the gastrodermis.

### 3.4. Layer-Specific Modulations of Wnt3 Expression Induced by ALP Treatment

In parallel, we tested the putative layer-specific modulations of *Wnt3* by exposing it to ALP animals of the epidermal and gastrodermal *HyAct-1388*:mCherry_*HyWnt3-2149*:GFP transgenic lines (named *HyWnt3-2149*:GFP), where GFP expression is under the control of 2149 bp of the *Wnt3* promoter [27,33] (Appendix A). With regard to *Wnt3* regulation after a two-day ALP treatment, the epidermal *HyWnt3-2149*:GFP persists at the tip of the hypostome while being strongly up-regulated in the tentacle ring and in tentacle roots, whereas small dots expressing *Wnt3* and *HyWnt3-2149*:GFP become visible along the body column. After a four-day ALP treatment, as expected, *Sp5* is strongly down-regulated, while a dense network of *Wnt3* or *HyWnt3-2149*:GFP dots is established along the entire body column in both layers (Figure 3B and Appendix A). We also noted a strong overall increase in gastrodermal *HyWnt3-2149*:GFP expression along the body column, indicating that the transactivation driven by the *HyWnt3-2149* promoter in the gastrodermis is much greater than that driven by the full set of regulatory sequences of the endogenous *HyWnt3* gene.

In summary, the analysis of *GFP* expression in these four transgenic lines helps identify the layer-specific regulation of *Sp5* and *Wnt3* in response to the ALP-induced activation of Wnt/β-catenin signaling (Figure 3C). The monitoring of GFP fluorescence in *HySp5-3169*:GFP animals confirmed these layer-specific differences, i.e., a transient *Sp5* up-regulation in the body column after two or four days of ALP exposure, followed by a down-regulation when Wnt3/β-catenin signaling is highly active, mimicking the situation at the tip of the hypostome. Indeed, after a seven-day ALP treatment, *HySp5-3169*:GFP fluorescence is restricted to the modified apical and basal extremities in epidermal transgenic animals and shifted to the lower half of the body in gastrodermal ones (Figure 3D).

### 3.5. β-Catenin Knock-Down Differentially Impacts Sp5 Expression in Epidermis and Gastrodermis

To test a possible layer-specific regulation of *Sp5* when β-catenin signaling is reduced, we knocked-down *β-catenin* in *HySp5-3169*:GFP transgenic animals (Figure 4A). As early as 24 h after the 1st electroporation (EP), we found the normalized levels of *β-catenin* transcripts significantly decreased by about 2-fold in each layer of the apical region and by 2-fold along the gastrodermis of the body column. Surprisingly, in gastrodermal *HySp5-3169:*GFP animals exposed to scrambled siRNAs, *β-catenin* transcript levels increase steadily after each EP in the apical region and body column. This observation suggests that the EP procedure leads to an unspecific stress-induced response that either activates *β-catenin* regulatory sequences and/or stabilizes *β-catenin* transcripts (Figure 4B and Appendix A).

With regard to the *Sp5* levels, we did not detect any specific modulation, with the exception of a transient decrease one day post-EP2 along the body column of gastrodermal *HySp5-3169*:GFP animals (Figure 4B and Appendix A). Concerning the *GFP* levels, we recorded in the apical and body column regions of the epidermal *HySp5-3169*:GFP animals a progressive decrease below 25% in the apical region one day after EP3, a result that likely reflects the weaker activation of the *HySp5-3169* promoter in the epidermis when *β-catenin* expression is decreased. We also noted in animals exposed to scrambled siRNAs a twofold increase in the level of epidermal *GFP* transcripts along the body column, again pointing to an EP-induced stress response. The consequences of *β-catenin* knock-down are different in the gastrodermis and are actually quite limited with a transient increase in the *HySp5-3169*:*GFP* transcript level in the apical region one day after EP2 without any significant modulation in the body column. It should be noted that the non-specific EP-induced increase in *GFP* observed in the epidermis is not observed in the gastrodermis.

In summary, the *β-catenin* RNAi procedure efficiently reduces the level of *β-catenin* transcripts, up to twofold in both the epidermis (mainly apical) and the gastrodermis (apical and body column). This *β-catenin* reduction does not affect the levels of *Sp5* transcripts outside the −1.41/+ 1.41-fold range (except at one time-point in the gastrodermal body column). Notably, it strongly affects the *HySp5-3169*:*GFP* transcript levels in the epidermis where a two-to-four-fold reduction (apical and body column) is noted. Such an effect is not observed in the gastrodermis. Finally, we noted that the RNAi procedure produces an EP-induced stress response in the body column leading to an unspecific increase in *β-catenin* levels in the gastrodermis and in *HySp5-3169*:*GFP* levels in the epidermis.

### 3.6. β-catenin Knock-Down Leads to Formation of Bud-like Structures Expressing Gastrodermal Sp5

We previously showed that a transient knock-down of *β-catenin* triggers a size reduction in *Hv_Basel* animals, as well as the formation of “bud-like structures”, which grow from the body column similarly to buds, but without forming a complete head with a fully differentiated ring of tentacles [33]. These bud-like structures are present in 100% of *Hv_Basel* animals one day after EP3 (Appendix A). Remarkably, as early as two days after EP1, well-defined regions along the parental polyp already strongly express *Sp5*, even though the bud-like structures are not morphologically visible yet (Figure 4C and Appendix A).

Concerning GFP and mCherry fluorescence in epidermal_*HySp5-3169*:GFP animals, the control animals exposed to scramble siRNAs show the expected pattern of high-level GFP fluorescence in the apical region and a low one along the body column. Meanwhile, the *β-catenin*(RNAi) animals exhibit a loss of GFP fluorescence two days post-EP2 in well-defined areas of the apical region together with a globally reduced GFP fluorescence along the body column, except for some GFP-positive patches (Figure 4D and Appendix A). Three days post-EP3, newly formed bud-like structures become visible in 30% to 60% of the animals and they never show any epidermal GFP fluorescence. We confirmed these findings by immuno-detecting GFP and mCherry in epidermal *HySp5-3169:*GFP animals knocked-down for *β-catenin*.

We noted the loss of epidermal GFP expression in large parts of the apical region, the presence of GFP-positive patches along the body column and the lack of GFP protein in the bud-like structures (Figure 4E and Appendix A). In gastrodermal *HySp5-3169*:GFP animals knocked-down for *β-catenin*, the gastrodermal layer remains GFP fluorescent in the tentacle ring and along the body column. The bud-like structures are all GFP fluorescent, with a positive signal in the presumptive apical region and often with a patchy pattern (Figure 4D and Appendix A). We confirmed these findings by immunodetecting GFP and mCherry six days post-EP2 in these animals (Figure 4E and Appendix A).

In summary, *β-catenin*(RNAi) leads to the formation of bud-like structures in both *Hv_Basel* and *Hv_AEP2* animals, a phenotype observed with a higher penetrance in the former (100% post-EP3) than in the latter (27% to 60%). These bud-like structures induced by *β-catenin*(RNAi) show low *HySp5-3169:GFP*/GFP expression in the epidermis, but high in the gastrodermis, in contrast to what is observed in natural buds (Figure 2E–H). This localized high level of Sp5 might explain why bud-like structures do not differentiate hypostome or tentacle rings. These results again point to a differential regulation of *HySp5-3169:GFP* by β-catenin signaling in the epidermis and gastrodermis.

### 3.7. Negative Auto-Regulation of Sp5 in the Epidermis

To determine whether the transcription factor Sp5 regulates its own expression, we knocked-down *Sp5* in *HySp5-3169*:GFP transgenic animals and monitored in each layer changes in *GFP*, *Wnt3* and *Sp5* expression as well as changes in GFP fluorescence at different time points after EP1 and EP2. We anticipated that after *Sp5*(RNAi), *GFP*/GFP expression would be increased if Sp5 autoregulation was negative and decreased if Sp5 autoregulation was positive. As previously reported, we, however, noticed some unspecific EP-induced increases in the transcript levels in animals exposed to scramble siRNAs. Here, this corresponds to an unspecific increase in *Wnt3* levels in both layers, maximal in the body column at 8 h post-EP1 and post-EP2. For *Sp5*(RNAi), we found in epidermal *HySp5-3169*:*GFP* animals, GFP transcripts more abundant 16 and 24 h after EP1 and EP2, two-to-four-fold in the apical region and above four-fold in the body column (Figure 5A and Appendix A). This GFP modulation was not observed in gastrodermal *HySp5-3169*:*GFP* animals. Regarding *Wnt3* and *Sp5* transcript levels, we did not detect any significant modulation by qPCR analysis, neither in the epidermal nor in the gastrodermal line. 

The analysis of the *GFP* expression pattern in *Sp5*(RNAi) epidermal *HySp5-3169*:GFP transgenic animals confirmed the above results, with an increase in *GFP* levels at the same time-points (Figure 5B and Appendix A). At the protein level, we first noted at 8 h post-EP1 some weak epidermal GFP fluorescence along the body column of some control and *Sp5*(RNAi) animals, possibly linked to the EP-induced activation of the *HySp5-3169* promoter. At 16 h post-EP1, we recorded a marked and specific increase in epidermal GFP fluorescence along the body column of *Sp5*(RNAi) animals, which is maintained high up to 24 h post-EP2 (Figure 5C and Appendix A). However, at two days post-EP2, when the ectopic epidermal GFP fluorescence is still detected, we found the *GFP* transcript levels in *Sp5*(RNAi) epidermal *HySp5-3169*:GFP transgenic animals significantly reduced in the apical and body column regions, implying that the *Sp5*(RNAi)-induced up-regulation of *HySp5-3169:GFP* is transient (Figure 5D). In gastrodermal *HySp5-3169*:GFP animals, we did not detect the global or localized modulation of *GFP* expression after *Sp5*(RNAi) (Figure 5A,B and Appendix A). We also did not record any significant change in GFP fluorescence after EP1 (Figure 5C and Appendix A). However, after EP2, we noted, in about half of the animals, ectopic spots of GFP fluorescence in the lowest part of body column or in the tentacles, indicating that *Sp5-*negative autoregulation might also take place in the gastrodermis, albeit more spatially restricted than in the epidermis.

### 3.8. Up-regulation of β-catenin after Sp5(RNAi) in Epidermis and Gastrodermis

We also performed concomitant q-PCR analysis of *Sp5*, *Wnt3* and *β-catenin* transcript levels in *HySp5-3169*:GFP animals knocked-down for *Sp5* at two days post-EP2. In epidermal *HySp5-3169*:GFP animals, we detected a significant increase in *β-catenin* transcripts in the body column and basal regions in the absence of significant modulations of *Sp5* and *Wnt3* levels (Figure 5D). In gastrodermal *HySp5-3169*:GFP animals, we likewise noted a significant increase in the levels of *β-catenin* transcripts in these two regions, alongside a slight reduction in the *Sp5* and *GFP* transcript levels (Figure 5D). This up-regulation of *β-catenin* upon *Sp5*(RNAi), which is detected along the body column to a similar extent in both layers, is expected since the negative regulation played by Sp5 on *Wnt/β-catenin* expression is reduced when *Sp5* is down-regulated, even transiently. Such regulation is, however, not detected in the apical region, consistently with previous results [33].

Despite the lack of the sustained down-regulation of *Sp5* transcripts after *Sp5*(RNAi), we conclude that the two-step RNAi procedure we applied is effective, highlighting the dynamic regulation of *Sp5* in each epithelial layer, with a 24 h-long up-regulation of *HySp5-3169*:*GFP* along the epidermis after each exposure to *Sp5*(RNAi). We did not record such a modulation of *HySp5-3169*:*GFP* in the gastrodermis. We interpreted the up-regulation of the epidermal *GFP* transcripts after *Sp5*(RNAi) as the consequence of knocking down the negative autoregulation played by the Sp5 transcription factor on its own expression. This is, however, only transient, as one day later, the *GFP* expression levels decreased by about twofold, probably as a consequence of the up-regulation of *β-catenin* expression that takes place in both layers of the body column, leading to a transient up-regulation of *Sp5* between 24 and 48 h after EP2, producing the Sp5 protein at a level where it represses the *Sp5* promoter and hence decreases the *GFP* transcript levels. However, two days after EP2, an ectopic GFP fluorescence was still visible in the epidermis (Figure 5E and Appendix A), in keeping with the long lifespan of the GFP protein [43].

### 3.9. Identification of Five Active Sp5-Binding Sites within the Proximal Hydra Sp5 Promoter

The *Hydra* Sp5 transcription factor belongs to the Sp/KLF family, a class of DNA-binding proteins that bind GC-rich boxes or GT/CACC elements through their three zinc finger (ZF) domains [32,44]. We previously identified, by ChIP-seq analysis performed with extracts from HEK293T cells, five Sp5-binding sites (Sp5-BS) and five TCF-binding sites (TCF-BS) within 2966 bp of the *Sp5* promoter, clustered in two adjacent regions named PPA and PPB in the vicinity of the *Sp5* transcriptional start site [33]. To identify active Sp5-binding sites (Sp5-BS) in *Hydra Sp5* promoter sequences, we raised antibodies against the *Hydra* Sp5 protein with the aim of performing a ChIP-qPCR analysis of the *HySp5* promoter using *Hydra* extracts and comparing Sp5-binding sites with those previously identified in human cells expressing HySp5. We raised two antibodies against *Hy*Sp5, one monoclonal and the other polyclonal, designed to target regions that do not contain evolutionarily conserved domains present in HySp5, such as the Sp box, the Buttonhead box and the ZF DNA-binding domain (Figure 6A and Appendix A). The two Sp5 antibodies specifically recognize the HySp5 protein, either as the HySp5-218 recombinant protein (24.5 kDA) used to raise the Sp5 monoclonal antibody, produced full length in vitro with the TNT reticulocyte transcription-coupled-translation system or expressed in transfected HEK293T cells (Figure 6B and Appendix A).

In the *Hv_AEP2* extracts, the monoclonal anti-Sp5 detects the Sp5 protein at higher levels in the apical region than in the body column as expected. In contrast, in three independent experiments, the polyclonal anti-Sp5 antibody recognizes a band at the appropriate size, but exclusively in nuclear extracts (NEs) prepared from the lower body column and not from the apical region (Appendix A). To evidence the possible cross-reactivity with the closely related Sp4 protein, we tested the polyclonal α-Sp5 antibody on the TNT-produced *Hydra* Sp4 protein but did not detect any band. We suspect that the polyclonal α-Sp5 antibody detects the Sp5 protein, but also cross-reacts with an unidentified Sp/KLF protein predominantly expressed in the body column and basal half of *Hydra*.

Next, we tested both the monoclonal and the polyclonal α-Sp5 antibodies in a ChIP-seq analysis of the Sp5-bound regions. We first used *Hm-105* extracts to assay the amplification of 15 regions along the 3169 bp of the *Sp5* promoter and 5′UTR sequences after ChIP (Figure 6C and Appendix A). Of these 15 regions, we found only two 100 bp-long regions specifically enriched with either antibody, but not by pre-immune serum, namely the overlapping PP4 (−135/−36) and PP5 (−71/+29) regions located in the proximal promoter of *Sp5*. Interestingly, the enrichment of the regions PP4 and PP5 by ChIP-qPCR is similar when extracts from *Hv_AEP* animals are used (Figure 6D and Appendix A). Moreover, these two regions were also identified when the ChIP-qPCR analysis was performed with extracts from human cells expressing HySp5 [33]. Each region contains three putative Sp5-BS, one of them (Sp5-BS3) being present in both PP4 and in PP5 (Figure 6E,F).

To test whether these putative Sp5-BS are functional in *Hydra*, we designed two double-stranded oligonucleotides (ds-DNAs) to perform an Electro-Mobility Shift Assay (EMSA) with (1) PPA (−135 to −67), which encompasses the identical Sp5-BS1 and Sp5-BS2 motifs, and (2) PPB (−71 to +2), which contains the distinct Sp5-BS3 and Sp5-BS4 motifs (Figure 6E–G). When *Hydra* NEs were incubated with biotin-labeled Sp5 ds-DNAs, we recorded a mobility shift, with two retarded bands for PPA and two distinct bands for PPB, no longer visible in the presence of a 200 fold excess of unlabeled oligonucleotides (Figure 6G). The mutation of Sp5-BS1 and Sp5-BS2 in the PPA region (C**CG**CCT -> C**TT**CCT) did not cancel the shift, but rather accentuated it. In contrast, when Sp5-BS3 (G**CG**CCA -> G**TT**CCA) and Sp5-BS4 (A**GG**CGT -> A**TT**CGT) in the PPB region were mutated, the shift almost disappeared. We, therefore, concluded that HySp5 likely binds the putative Sp5 binding sites in the PPA and PPB regions, with higher specificity in the latter; these results support the hypothesis that in *Hydra*, the Sp5 transcription factor is involved in *Sp5* autoregulation.

### 3.10. The Sp5 Proximal Promoter Is Involved in Sp5-Negative Autoregulation

To determine whether some of these five proximal Sp5-binding sites are indeed involved in Sp5 auto-regulation, we tested these sequences in an ex vivo transactivation assay system (Figure 7A). We prepared seven reporter constructs where the expression of luciferase is driven either by the full *HySp5* promoter (*HySp5*-2992:luciferase), by a shorter version where 164 bp of the proximal sequences are deleted (*HySp5*-2828:luciferase) or by the full *HySp5* promoter where one of the five *HySp5-BS* is mutated (*HySp5*-2992-mBS1:luciferase, -mBS2, -mBS3, -mBS4 and -mBS5). Each of these reporter constructs were co-expressed in HEK293T cells either with the full Sp5 protein under the control of the CMV promoter (CMV:HySp5-420) or with a truncated version of the Sp5 protein lacking the DNA-binding domain (CMV:HySp5-ΔDBD).

In conditions where Sp5 is either not expressed or expressed but in its truncated version (*Hy*Sp5-ΔDBD), we recorded low luciferase activity, consistent with the fact that the *HySp5-2992* promoter is poorly active in HEK293T cells (Vogg et al., 2019) [33]. In contrast, in the presence of the *Hy*Sp5-420 protein, we measured a 7 fold higher level of activity of the *HySp5-2992* promoter and obtained similar levels when the *HySp5-BS1*, *HySp5-BS2* and *HySp5-BS4* sites were mutated (Figure 7A). However, surprisingly, when the proximal sequences are completely deleted, the transactivation levels are more than doubled, indicating that these sequences actively repress the activity of the *HySp5*-2992 promoter in the presence of the *Hy*Sp5-420 protein. When the *HySp5-BS3* and *HySp5-BS5* sites are mutated, the activity is lower than that recorded when the *HySp5*-2992 promoter is complete, indicating that either these two sites play a positive role for the full activity of the *HySp5* promoter or they restrict the repressive activity of the proximal sequences. These results show that the *HySp5* promoter is subject to complex regulation, with a clear Sp5-dependent repressive role of the proximal sequences and an enhancing role of the more upstream ones.

### 3.11. The Zic4 Transcription Factor Positively Regulates Sp5 Expression

We recently showed that the *Hydra* transcription factor Zic4 (HyZic4), which is involved in the differentiation of tentacles and the maintenance of their identity, is a downstream target gene of *HySp5* [34]. We considered the possibility that Zic4 also regulates *Sp5* expression in a feedback loop. We first searched for the presence of Zic-binding sequences (Zic-BS) as deduced from those identified in vertebrate or non-vertebrate gene promoters [45] (Table 1). We identified two putative Zic-BS in the *HySp5-2992* upstream sequences at positions −670/−659 and −391/−369 (Table 1, Appendix A), as well two putative Zic-BS in the *HyWnt3-2149* upstream sequences and five in the *HyZic4* ones (Table 1, Appendix A). To test whether *Hy*Zic4 regulates *HySp5*, we expressed in HEK293T cells the *HySp5*-2992:luciferase construct together with either the full *Hy*Zic4 protein (CMV:Zic4-431) or a truncated form lacking its DNA-binding domain (*CMV:Hy*Zic4-ΔDBD) (Figure 7B). In the presence of *Hy*Zic4, the *HySp5*-2992:luciferase activity is multiplied by almost 50 fold when human β-catenin is co-expressed and over 50 fold when human β-catenin is not co-expressed. Remarkably, the luciferase activity becomes basal when the Zic4 DNA-binding domain is deleted, indicating that *Hy*Zic4 can enhance *HySp5* expression through directly binding to its promoter, independently of β-catenin.

Next, we tested the level of *HyZic4* promoter activity in HEK293T cells, which contain two putative Sp5-binding sites [34], when Sp5 is co-expressed. We measured a 10 fold increase in the *HyZic4*-3505:luciferase activity when the full Sp5 protein (*CMV*:*Hy*Sp5-420) is co-expressed, an increase no longer detected when the Sp5 protein lacks its DNA-binding domain (*CMV:Hy*Sp5-ΔDBD) (Figure 7C). As stated previously, this increase is observed at similar levels in the presence or absence of human β-catenin co-expression. This result indicates that Sp5 can significantly enhance *HyZic4* expression through direct DNA-binding.

Similarly, we also measured a strong Zic4 auto-activation (~50 fold), which requires the DNA-binding domain (Figure 7G). By contrast, when we used the TOPFlash assay where six tandem TCF-binding sites can enhance luciferase expression [61], we found that the TOPFLASH luciferase activity decreased when Zic4 is co-expressed. This Zic4-dependent repression requires the Zic4 DNA-binding domain, proved to be Zic4 dose-dependent and is enhanced when human β-catenin is co-expressed (Figure 7E,F). However, when we tested the Zic4 activity on the *Wnt3* promoter (2142 bp), we found a 10 fold increase in the *HyWnt3*-2142:luciferase activity, likely through a direct interaction as this increase is no longer observed when the Zic4 DNA-binding domain is deleted (Figure 7G).

We have summarized these interactions with those previously identified with Sp5 or Zic4 in HEK293T cells [28,29] in a scheme that presents a series of positive loops where Zic4 appears very potent, including in its own autoregulatory loop, and two negative regulations, with Sp5 repressing *Wnt3* expression and Zic4 repressing TCF-regulated promoters (Figure 7H). In HEK293T cells overexpressing *HyZic4*, we also show by co-immunoprecipitation that the two transcription factors HyZic4 and human TCF1 can physically interact (Figure 7I), similarly to what we previously showed between HySp5 and TCF1 [33]. These results suggest that in *Hydra* cells where Sp5 and Zic4 are co-expressed, they enhance each other’s expression while repressing TCF/β-catenin transcriptional activity.

We tested this gene regulatory network (GRN) in *Hydra* and indeed found in animals knocked-down for *Zic4* or *β-catenin* the *Zic4* levels down-regulated at least two-fold and also reduced after *Sp5* knock-down, but at a lower level (Figure 7J). We also found the levels of *Sp5* moderately down-regulated in animals knocked-down for *Zic4* or for *β-catenin* (Figure 7K). We concluded that the regulatory events recorded in HEK293T cells likely take place in *Hydra* cells where all components of this GRN are expressed, namely in the gastrodermal epithelial cells of the head region where *Sp5* and *Wnt3* are highly expressed and in the epithelial battery cells located in the epidermis of the tentacles (Figure 8A).

## 4. Discussion

### 4.1. Epithelial Layer-Specific Regulations of Sp5 in Intact Animals

To better decipher the dynamics of the interactions between the activating and inhibiting components of the head organizer in intact and regenerating *Hydra*, we compared the in vivo regulation of *Sp5* in each epithelial layer, epidermal and gastrodermal, when developmental, pharmacological or genetic conditions vary (see Table A1). For this purpose, we used transgenic lines expressing the reporter construct *HySp5-3169*:GFP either in the epidermis or gastrodermis and analyzed GFP expression in intact or developing animals, either regenerating or budding. In intact animals, we found several differences between epidermal and gastrodermal *GFP*/GFP expression; in the epidermis, *GFP*/GFP is expressed throughout the hypostome, while the tip of the hypostome is free of gastrodermal *GFP*/GFP, in agreement with the fact that *Sp5* transcripts are not detected in this area [33]. In addition, epidermal *GFP*/GFP is maximal in the tentacle ring and uppest body column, while the gastrodermal one extends along the body axis. In animals regenerating their heads, we noted a differential temporal regulation of *Sp5* in each layer, with *GFP* up-regulated during the early phase in the gastrodermis but one day later in the epidermis.

These layer-specific regulations of *HySp5-3169*:*GFP* were identified through different approaches: at the protein level, by monitoring in vivo GFP fluorescence and immunodetecting GFP protein expression, and at the transcript level, by performing qPCR and in situ hybridization to quantify and map endogeneous *Sp5* and *GFP* expression along the body axis. In addition, these layer-specific regulations of *Sp5* in intact animals are supported by the single-cell analysis of *Sp5* expression [35] that shows a predominant expression of *Sp5* in epithelial cells, with maximal levels observed in tentacle battery cells located in the epidermis, a lower level in epithelial cells of the hypostome and no expression in stem cells along the body column. In the gastrodermis, single-cell analysis detects the highest levels of *Sp5* in apical cells and in one sub-population of epithelial stem cells along the body column (Figure 8A). Therefore, we concluded that the *HySp5-3169*:*GFP* transgenic lines provide suitable and reliable tools to monitor the endogenous *Sp5* regulation, with the 3 kb of *Sp5* upstream sequences inserted in this construct being sufficient to direct *GFP* expression in a way that mimics endogenous *Sp5* regulation in each layer.

### 4.2. Three Architectures of the Wnt3/β-Catenin/TCF/Sp5/Zic4 GRN in Intact Hydra

This study adds a new level of understanding of how the head organizer works in *Hydra*. The parallel analysis of the *Wnt3/β-catenin/TCF/Sp5/Zic4* GRN in human HEK293T cells and in *Hydra* epithelial layers reveals complex cross-regulatory interactions. We first could confirm several positive regulations, those of Wnt3/β-catenin/TCF on *Sp5* and *Zic4* expression and that of Sp5 on *Zic4*, and identify some new ones such as Zic4 on *Sp5* and Zic4 on *Wnt3* (at least in HEK293T cells). We also confirmed the down-regulation of *Wnt3* by Sp5 and reported, as a new finding, the down-regulation of *β-catenin* by Sp5, as well as the down-regulation of β-catenin/TCF activity by Zic4 (at least in HEK293T cells). Finally, we identified autoregulatory loops, positive for Sp5/*Sp5* and Zic4/*Zic4* in HEK293T cells, positive for Wnt3/*Wnt3* and β-catenin/*β-catenin* via TCF in *Hydra* and negative for Sp5/*Sp5* in the *Hydra* epidermis (see Table A1).

By analyzing these interactions along the body axis of intact animals, we could characterize three distinct organizations of the *Wnt3/β-catenin/TCF/Sp5/Zic4* GRN that correspond to three distinct patterning functions in three anatomical contexts (Figure 8). Indeed, in the analysis of *Wnt3*, *β-catenin*, *Sp5* and *GFP* regulation in epidermal and gastrodermal *HySp5-3169:eGFP* and *HyWnt3-2149:eGFP* transgenic lines after ALP treatment, *β-catenin*(RNAi) or *Sp5*(RNAi) show that *Sp5* is differentially regulated in the epidermis and gastrodermis (1) at the apex, (2) in the tentacle zone and (3) along the body column (Figure 8B–E). At the apex of intact animals, the tightly spatially restricted expression of *Sp5* in the gastrodermis is crucial for maintaining maximal levels of *Wnt3* expression at the tip of the head, where Wnt3/β-catenin/TCF signaling drives the constitutive head organizer activity leading to head maintenance. In the tentacle zone, the positive co-regulation of *Sp5* and *Zic4* in the epidermis of the tentacle zone is critical for tentacle formation with an unclear role for the gastrodermis. Along the body column, the high levels of *Sp5* expression and Sp5 activity in the gastrodermis, possibly through positive autoregulation, are critical for keeping Wnt3/β-catenin/TCF signaling low and preventing the formation of ectopic tentacles, bud-like structures or ectopic heads.

### 4.3. Sp5-Negative Autoregulation in the Epidermis

Sp5 was identified as a target and a regulator of the Wnt transcriptional program in vertebrates [28,29,30,31,32], notably as a repressor [30,62]. We also showed that *Hydra* or zebrafish Sp5 expressed in human cells acts as an evolutionarily conserved transcriptional repressor, including on transcriptional machinery and on *Sp* genes [33]. This new study confirms this finding as in *Hydra*, Sp5 negatively regulates its own expression in the epidermis, as evidenced by the transient up-regulation of *Sp5* after each exposure to *Sp5* siRNAs. However, the response to *Sp5*(RNAi) in *HySp5*-3169:GFP transgenic animals is highly asymmetrical between the epithelial layers as in the epidermis, there is a transient but massive increase in *GFP* expression and GFP fluorescence along the body axis, together with a limited increase in *Sp5* and *Wnt3* levels along the body column. In contrast, in the gastrodermis, we recorded no overall increase in the *GFP* transcript level, but only a few ectopic spots of *GFP* overexpression or GFP fluorescence in the peduncle region and tentacles.

We interpret this transient GFP/*GFP* up-regulation in the epidermis together with that of *Sp5,* although more limited, as a release of the negative auto-regulation played by Sp5 in this layer. This epidermal-specific *Sp5* negative autoregulation suffices to explain the lower constitutive *Sp5* expression recorded in this layer. As such, this constitutive asymmetry in *Sp5* expression also explains the more effective *Sp5*(RNAi) knock-down in the epidermis as *Sp5* levels are low and epithelial cells are highly accessible to electroporated siRNAs, whereas in the gastrodermis, *Sp5* levels are higher and epithelial cells are less accessible.

Among the 2992 bp *Sp5* upstream sequences, the ChIP-seq analysis could identify only two areas in the *Sp5* proximal promoter, PPA and PPB, which are enriched in the Sp5 protein when *Hm-105* or *Hv_AEP2* nuclear extracts are used. In addition, these two areas, which each contain two Sp5-binding sites previously identified by ChIP-seq analysis in human cells expressing HySp5, are necessary for Sp5-negative autoregulation. The role of Sp5-negative autoregulation in the GRN is further supported by these new findings, showing that Sp5 can directly regulate its own promoter.

*Sp5* regulation is quite dynamic, as observed in *HySp5*-3169:GFP animals exposed to *Sp5*(RNAi) where *GFP* levels are first found up-regulated in the epidermis, as discussed above, then one day later, significantly down-regulated in the apical region and body column, providing evidence for a return to Sp5-repressive activity. In parallel, the *Sp5* and *Wnt3* transcript levels remain unaffected, possibly as the result of the highly dynamic cross-regulations that take place between these genes. In the gastrodermis, the *Sp5* and *GFP* levels are only mildly decreased in response to *Sp5*(RNAi), whereas *β-catenin* is up-regulated in both layers along the body column indicating that in intact animals, Sp5 acts as a head inhibitor by repressing not only *Wnt3*, but also *β-catenin* expression, thus reinforcing the feed-back loop that reduces head activation, i.e., Wnt3/β-catenin signaling activity.

### 4.4. Distinct Configurations of the Wnt3/β-Catenin/Sp5 GRN in the Homeostatic and Developmental Head Organizers

This study shows that the temporal and spatial regulation of *Sp5* and *Wnt3* in each epithelial layer is in fact different in the homeostatic and developmental organizers. In intact animals, the absence of *HySp5-3169*:GFP expression in the gastrodermal epithelial cells of the perioral region where *Wnt3* expression is maximal indicates that a high level of Wnt/β-catenin signaling can only be stably maintained if *Sp5* expression is repressed. This equilibrium situation is necessary to maintain homeostatic organizer activity. How *Sp5* expression is maintained inhibited within gastrodermal cells of the homeostatic head organizer has not yet been identified. Since in each context where *Wnt3* expression is highest, *Sp5* is repressed, we infer that the highest levels of Wnt/β-catenin signaling induce the transcriptional repression of *Sp5* and/or degradation of *Sp5* transcripts.

In contrast, in the apical-regenerating tip, the *Sp5* expression domain is broad in the gastrodermis, likely necessary for restricting head organizer activity, i.e., limiting *Wnt3* and *β-catenin* up-regulation as well as the activation of Wnt3/β-catenin/TCF signaling, as such as a single head develops instead of multiple ones. The *Wnt3* and *Sp5* gastrodermal domains overlap at least for the first 24 h after a mid-gastric bisection, with *Sp5* expressed at a high level in these cells. Therefore, we believe that in gastrodermal epithelial cells that have the power to develop organizing activity, as in apical-regenerating tips, Wnt3/β-catenin signaling is protected from Sp5 activity, either by the active inhibition of Sp5 repressive transcriptional activity on *Wnt3* or *β-catenin* expression or by the enhanced degradation of the Sp5 protein. In conclusion, it remains to be understood how interactions between Wnt3/β-catenin signaling and the Sp5 transcription factor remain distinct in the homeostatic and developmental head organizers, possibly via transcriptional or post-transcriptional mechanisms in the former context and via translational or post-translational mechanisms in the latter one.

### 4.5. Phenotypic Changes Induced by Dysregulations of the Wnt3/β-Catenin/TCF/Sp5/Zic4 GRN

The perturbations of the dynamic interactions within the *Wnt3/β-catenin/TCF/Sp5/Zic4* GRN induce several phenotypes along the body axis such as (i) the loss of tentacle identity after *Zic4*(RNAi) [34], (ii) the formation of ectopic tentacles after ALP treatment [23] associated with an overall increase in gastrodermal *Sp5* expression and a punctuated up-regulation of *Wnt3* in both layers, (iii) the formation of multiple ectopic heads induced by *Sp5*(RNAi) linked to the time- and space-restricted decrease in gastrodermal *Sp5* associated with the localized activation of Wnt3/β-catenin/TCF signaling, as observed in *Hv_Basel* [33], and (iv) the formation of bud-like structures upon *β-catenin*(RNAi), which rarely differentiate a head. This latter phenotype is highly penetrant in *Hv_Basel*, was present in 100% of animals one day after EP3 and was identical but less penetrant in *Hv_AEP2* transgenic animals.

The regulations of *Sp5* in *HySp5*-3169:GFP transgenic animals knocked-down for *β-catenin* are layer-specific, consistent with the observed phenotype. In the epidermis, *β-catenin*(RNAi) leads to a drastic reduction in *GFP* expression and GFP fluorescence, consistent with the fact that *Sp5* is directly positively regulated by Wnt/β-catenin signaling [33]. In contrast, in the gastrodermis, the localized up-regulation of *GFP* expression and GFP fluorescence in the bud-like structures were unexpected. The preliminary results indicate that the RNAi-induced decrease in the *β-catenin* transcript levels leads to the rapid nuclear translocation of the β-catenin protein available in gastrodermal cells and the subsequent transactivation of β-catenin/TCF target genes such as *Sp5*.

This paradoxal response to *β-catenin*(RNAi) explains (a) the rapid growth of bud-like structures in starved animals that normally do not bud, (b) the high level of gastrodermal *HySp5-3169*:*GFP* and *Sp5* expression in these structures and (c) the lack of head structure differentiation due to the high level of Sp5 and the inhibition of the head organizer. This two-step response to *β-catenin* knock-down provides an experimental paradigm for inducing the proliferative phase of the budding process in the absence of apical differentiation and for characterizing the molecular players required in parental tissues.

### 4.6. Variability of Head Organizer Inhibitor Strength across Hydra Strains

In this study, we observed great phenotypic variability between *Hv_Basel* and *Hv_AEP2* after exposure to GRN modulators, whether ALP treatment or *Sp5* knockdown. After ALP, *Hv_Basel* developed numerous ectopic tentacles along their body column within a few days, whereas *Hv_AEP2* formed very few, even after seven days of ALP exposure. In *Hv_Basel*, a two-day exposure to ALP led to a first wave of *Sp5* up-regulation, followed two days later by a second wave of *Wnt3* up-regulation, inducing the formation of small *Wnt3* spots along the body column and the subsequent emergence of multiple ectopic tentacles, as observed in *Hv_ZüL2*.

In contrast, in *Hv_AEP2* animals, either non-transgenic or transgenic (*Sp5-3169*:GFP and *Wnt3-2149*:GFP animals), exposure to ALP led to two waves of gene regulation that were slightly different. In the first phase, large circles of cells transiently expressing *Sp5* were formed in the epidermis and a more diffuse and intense expression of *Sp5* was observed in the gastrodermis. This phase was followed by the appearance of *Wnt3* spots along the body column, with just a few in the epidermis, but multiple ones in the gastrodermis together with a *Wnt3* diffuse expression (Figure 8D,E). Nevertheless, only rare ectopic tentacles were formed in *Hv_AEP2* animals. We infer that despite multiple spots of high *Wnt3*, Sp5 activity is constitutively higher along the gastrodermis when compared to *Hv_Basel*, repressing *β-catenin* which is kept minimal in SC2 epithelial stem cells.

After *Sp5*(RNAi), all *Hv_Basel* animals become multiheaded, whereas *Hv_AEP2* animals treated in the same way do not. This highly penetrant multiheaded phenotype in *Hv_Basel* occurs without affecting the original head region, likely because *Sp5* expression remains high there, less subject to modulation and sufficient to have a phenotypic impact (Figure 8B,C). In contrast, in *Hv_Basel,* the body column acquires the properties of a head organizer after *Sp5*(RNAi). However, this does not happen in *Hv_AEP2* animals, where the absence of ectopic axis formation upon *Sp5(RNAi)* is explained by the fact that gastrodermal cells express *Sp5* at higher constitutive levels than in *Hv_Basel*, maintaining Sp5 repression on *Wnt3* and *β-catenin* expression and preventing ectopic axis formation.

In conclusion, the rarity of the ectopic tentacle phenotype after ALP or the absence of the multiple-head phenotype after exposure to *Sp5*(RNAi) in *Hv_AEP2* animals result from the stronger activity of the head organizer inhibitor along the body column in these animals compared to that present along the body column of *Hv_Basel* or *Hv_ZüL2* animals. These results are consistent with previous studies that revealed, through systematic transplantation experiments performed on a variety of *Hydra* strains, significant variations in the respective strengths of the head activation and head inhibition components along the apical-to-basal axis between *Hydra* strains [7,63,64]. These results also highlight the predominant role of the gastrodermis in the negative regulation of the head organizer as previously demonstrated by producing chimeric animals with gastrodermal epithelial cells isolated from strains with low or high levels of head inhibition [65].

## 5. Conclusions

This study, based on the analysis of *Sp5* and *Wnt3* regulation in each epithelial layer of *Hydra*, reveals distinct architectures of the *Wnt3/β-catenin/TCF/Sp5/Zic4* GRN in different anatomical regions of the animal. Each architecture is characterized by a specific relative weight for each component, providing a unique combination that controls or prevents a specific patterning process. In the context of tentacle formation, the β-catenin-dependent activation of a subset of the GRN, namely Sp5 and Zic4, plays the leading role in the epidermis and the head activator component is kept inactive. In the context of head maintenance or head formation, the head activator component, i.e., Wnt3/β-catenin/TCF signaling, plays the leading role in the gastrodermis. In the context of the body column, the head inhibitor component plays the leading role in the gastrodermis to keep the head organizer locked, even though the time- and space-restricted down-regulation of *Sp5* can occur, supporting the localized activation of Wnt3/β-catenin/TCF signaling and further ectopic head formation.

The next step will be to compare the epidermal and gastrodermal chromatin signatures in the hypostome, the apical-regenerating tips and the tentacle ring along the body column to map the regulatory sites linked to the context-specific GRN architecture. The complete set of actors involved in each context as well as the role they play in relation to the *Wnt3/β-catenin/TCF/Sp5/Zic4* GRN remains to be identified, e.g., Brachyury [66,67] and MAPK/CREB [24,41,68,69,70] for head activation; notum [66,67], Dkk1/2/4, Thrombospondin and HAS7 [71,72,73,74] for head inhibition; and *Alx* and Notch signaling for tentacle formation [75,76,77]. Also, further questions to investigate will be how this GRN, which is conserved across evolution [32,33,62], moves from one architecture to another, typically when Sp5 or β-catenin levels reach some threshold values or when additional players modify the GRN patterning function. More generally, an in-depth understanding of the molecular mechanisms behind the formation of the organizing center should enable us to transform a somatic tissue into one endowed with developmental organizing properties.

## Figures and Tables

**Figure 2 biomedicines-12-01274-f002:**
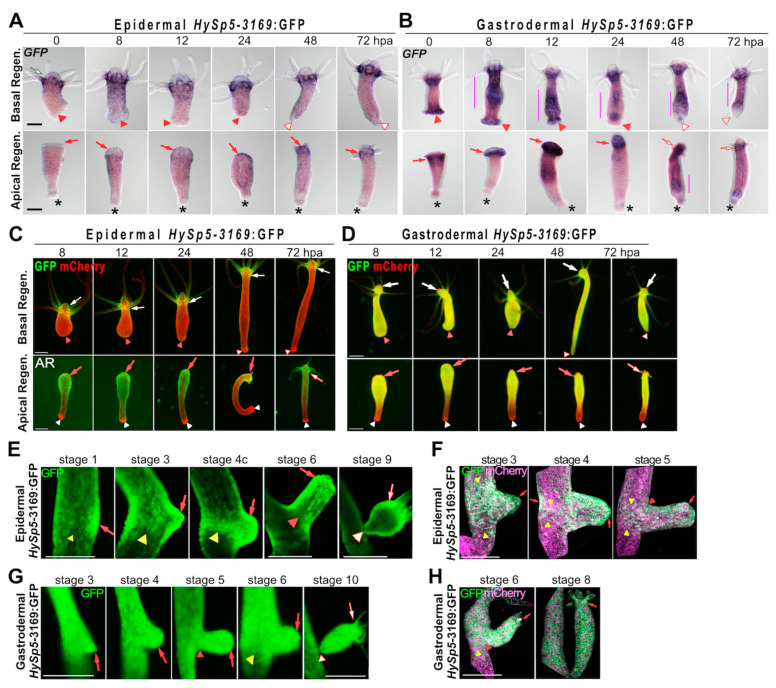
*GFP* regulation in regenerating and budding *HySp5-3169*:GFP transgenic animals. (**A**,**B**) *GFP* expression in regenerating halves from epidermal (**A**) and gastrodermal (**B**) *HySp5-3169*:GFP transgenic animals bisected at t0 and fixed at indicated times. Regen.: regeneration; hpa: hours post-amputation; red arrows point to apical-regenerating (AR) regions, red triangles to the basal-regenerating (BR) regions, vertical bars indicate gastrodermal *GFP* expression along the body column, asterisks the original basal discs, white arrows outlined red to the regenerated heads, white triangles outlined red to the regenerated basal discs. See Appendix A. (**C**,**D**) GFP (green) and mCherry (red) fluorescence in AR and BR halves of *HySp5-3169*:GFP transgenic animals pictured live at indicated time points. White arrows point to apical regions of original polyps, red arrows to AR regions, white arrows outlined red to regenerated heads; white arrowheads to original mature basal discs, red arrowheads to the BR regions. See Appendix A. (**E**–**H**) Live imaging of budding *HySp5-3961*:GFP transgenic animals, either epidermal (**E**) or gastrodermal (**F**), pictured at indicated stages with the Olympus SZX10 microscope ((**E**,**G**), GFP fluorescence only) or the Zeiss LSM780 microscope ((**F**,**H**), GFP and mCherry fluorescence). On the parental polyp, yellow arrowheads point to the “budding belt” that forms in the budding zone; on the developing buds, red arrows point to the developing apical region, red arrowheads to the differentiating basal region and white arrowheads outlined red to fully differentiated basal discs. Scale bar: 250 µm.

**Figure 3 biomedicines-12-01274-f003:**
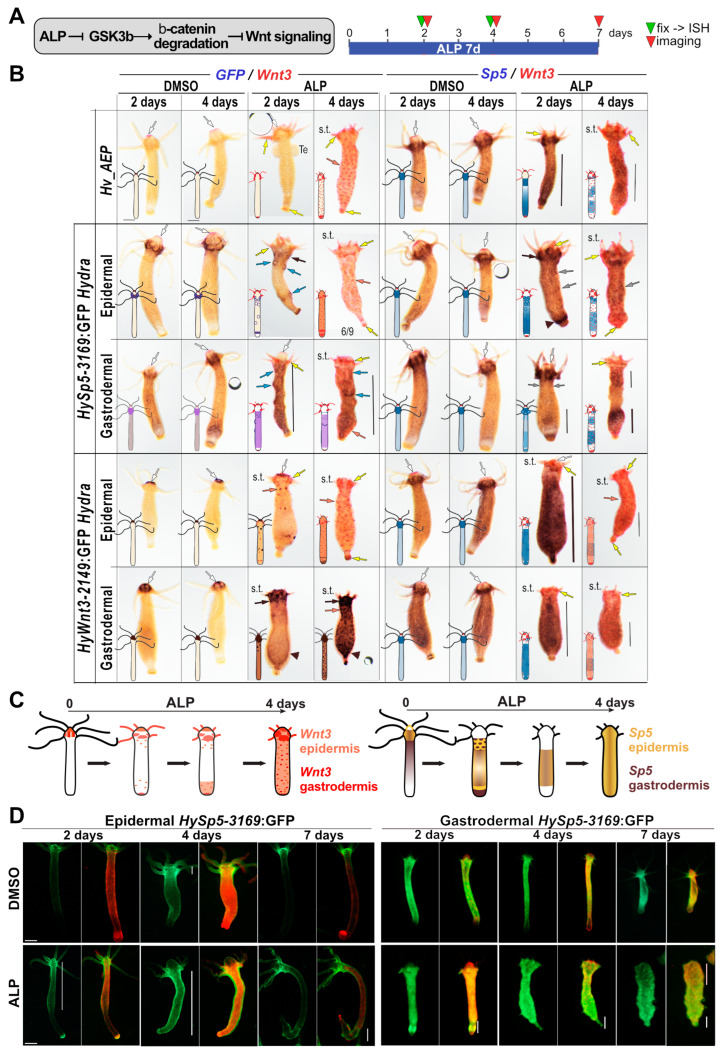
Alsterpaullone-induced modulations of *GFP*, *Wnt3* and *Sp5* expression along the epidermis and gastrodermis of *HySp5-3169*:GFP and *HyWnt3-2149*:GFP transgenic animals. (**A**) Schematic representation of the activating effect of ALP on Wnt/β-catenin signaling. (**B**) Co-detection of *GFP* (purple) and *Wnt3* (red) (left half) or *Sp5* (purple) and *Wnt3* (red) (right half) in wild-type *Hv_AEP* animals or in transgenic animals that constitutively express the *HySp5*-3169:GFP or *HyWnt3*-2149:GFP constructs, after 2- or 4-day ALP exposure. White arrows: *Wnt3* expression at the tip of the hypostome; black arrows: expression immediately below the apical region; black arrowheads: ALP-induced *GFP* expression in the peduncle zone; blue and grey arrows: ALP-induced circular zones of *GFP* and *Sp5* expression respectively along the body column; yellow arrows: ALP-induced ectopic *Wnt3* expression in the apical or basal regions; orange arrows: *Wnt3*-expressing spots along the body column; vertical black bars: areas of ALP-induced *GFP* expression along the body column; s.t.: short tentacles; Te: testis. (**C**) Schematic representation of the ALP-induced modulations of *Wnt3* and *Sp5* in the epidermal and gastrodermal *HySp5-3169:GFP* and *HyWnt3-2149:GFP* transgenic lines. See Appendix A. (**D**) Live imaging of mCherry and GFP fluorescence in epidermal and gastrodermal *HySp5*-3169:GFP animals treated for 2, 4 and 7 days with ALP or DMSO. For each condition, GFP fluorescence is shown on the left and the merged GFP (green) and mCherry (red) fluorescence on the right. Vertical white bars indicate areas of ectopic GFP fluorescence. Scale bar: 250 µm.

**Figure 4 biomedicines-12-01274-f004:**
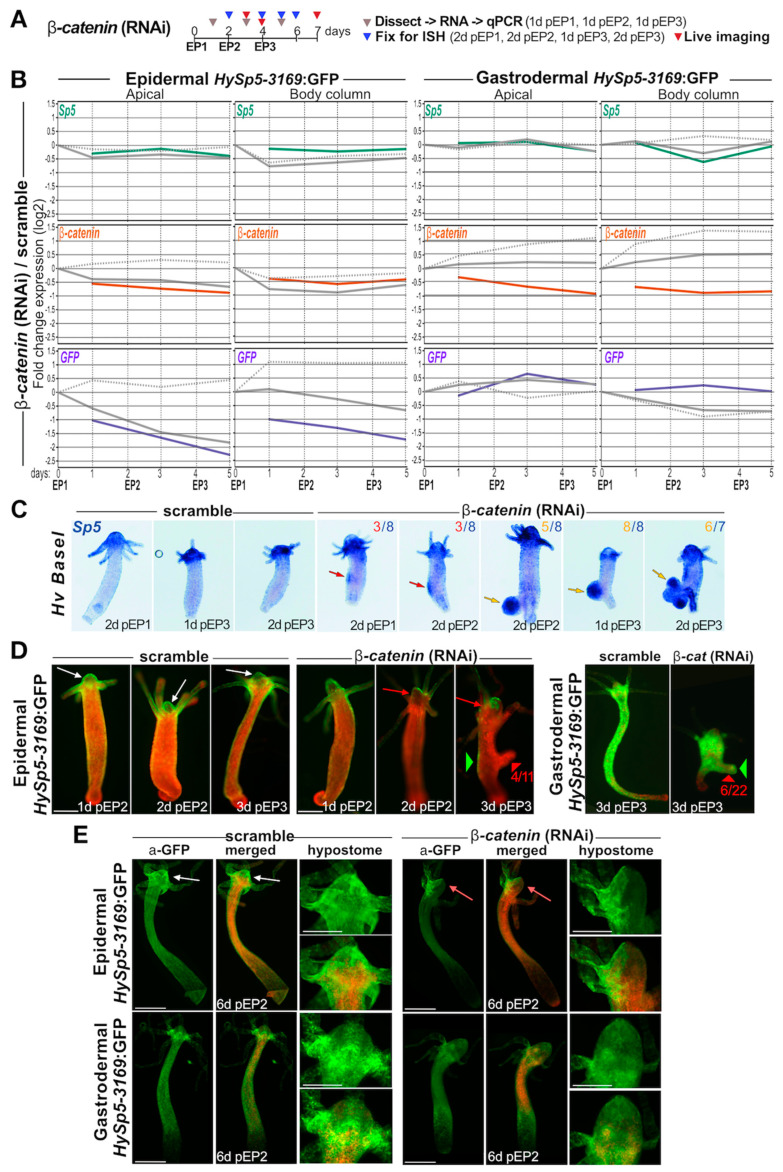
Impact of *β-catenin*(RNAi) on *Sp5* expression in *Hv-Basel* and *HySp5-3169*:GFP transgenic animals. (**A**) Schematic view of the procedure: After one, two or three electroporations (EP1, EP2 and EP3) with scramble or *β-catenin* siRNAs, animals were either dissected in apical and body column regions for RNA extraction (grey triangles), or fixed for whole-mount in situ hybridization (ISH, blue triangles) or imaged live (red triangles) at indicated time-points (d pEP: day(s) post-EP). (**B**) Q-PCR analysis of *Sp5*, *β-catenin* and *GFP* transcript levels in apical (100–80%, left) and body column (80–0%, right) regions of epidermal (left) and gastrodermal (right) *HySp5-3169*:GFP transgenic animals taken one or two days after EP1 (days 1 and 2), after EP2 (days 3 and 4) or one day after EP3 (day 5). In each panel, the colored line corresponds to the Fold Change (FC) values between *β-catenin* RNAi animals (continuous grey lines) and control animals exposed to scramble siRNAs (dotted grey lines). Values are each expressed as FC relative to non-electroporated animals at time 0, just before EP1. See Appendix A. (**C**) *Sp5* expression pattern and phenotypic changes in scramble and *β-catenin*(RNAi) *Hv_Basel* animals at indicated time-points. Red arrows point to *Sp5*-expressing patches along the body column, yellow arrows to bud-like structures that express *Sp5* and become multiple two days pEP3 without differentiating apical structures. See Appendix A. (**D**) GFP (green) and mCherry (red) fluorescence in *β-catenin*(RNAi) *HySp5-3169*:GFP transgenic live animals. GFP fluorescence in apical regions is either normal (white arrows) or missing (red arrows). Note the transient bud-like structures that develop after *β-catenin*(RNAi) (red arrowheads) and express GFP in gastrodermal_*HySp5-3169*:GFP animals (green triangles). See Appendix A. (**E**) Immunodetection of GFP and mCherry in *β-catenin*(RNAi) *HySp5*-*3169*:GFP animals 6 days post-EP2 (6d pEP2). Apical regions of scramble and *β-catenin*(RNAi) animals are magnified on the right. White and red arrows as in (**D**). See Appendix A. Scale bar: 250 µm.

**Figure 5 biomedicines-12-01274-f005:**
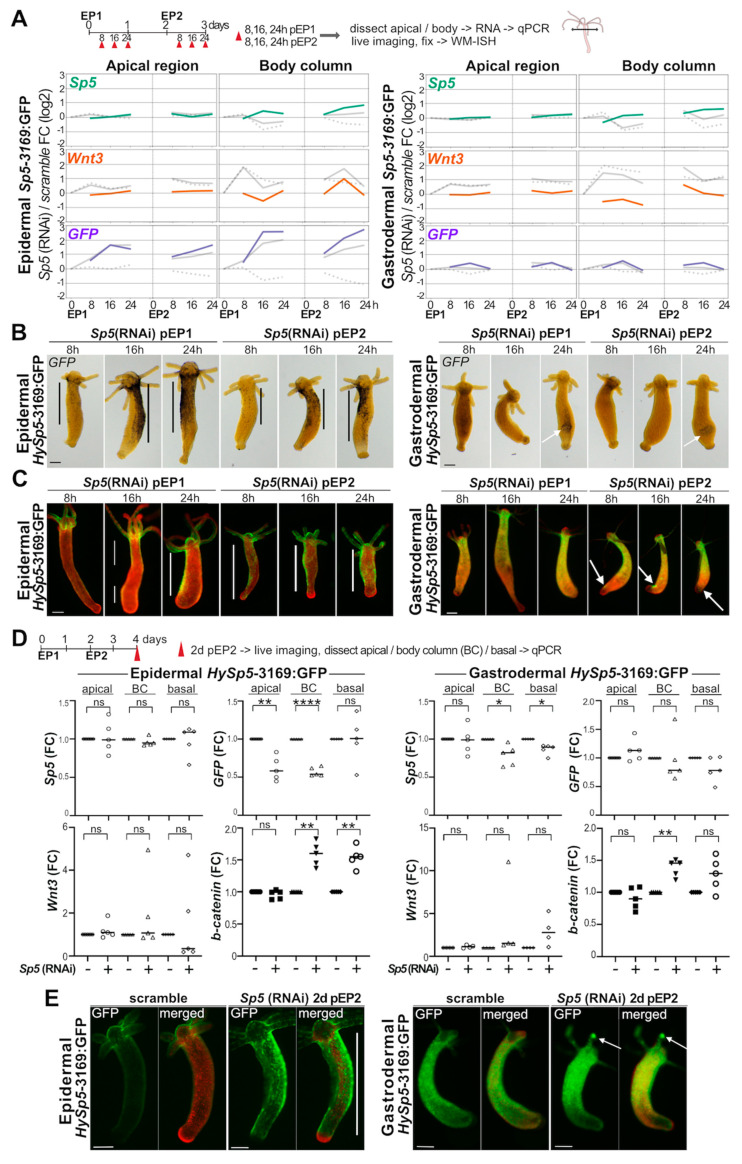
Ectopic *GFP*/GFP expression in *HySp5-3169*:GFP animals knocked-down for *Sp5*. (**A**) RNAi procedure applied in experiments depicted in panels A-C. At 8 h, 16 h and 24 h post-EP1 (pEP1) and 8 h, 16 h and 24 h post-EP2 (pEP2, red triangles), animals were either fixed for RNA extraction, or imaged live and fixed for whole-mount ISH. Q-PCR analysis of *Sp5*, *β-catenin* and *GFP* transcript levels in apical (100–80%, left) and body column (80–0%, right) regions of epidermal (left) and gastrodermal (right) *HySp5-3169*:GFP transgenic animals exposed to scramble siRNAs or to *Sp5* siRNAs. In each panel, the colored line corresponds to the Fold Change (FC) values between *Sp5* RNAi animals (continuous grey lines) and control animals exposed to scramble siRNAs (dotted grey lines), which are each expressed as FC relative to non-electroporated animals at time 0, just before EP1. See Appendix A. (**B**) *GFP* expression detected by WM-ISH at indicated time-points after *Sp5*(RNAi), as depicted in (**A**). Vertical black bars along the body column and white arrows in the lower body column indicate regions where *GFP* is up-regulated. See Appendix A. (**C**) GFP (green) and mCherry (red) fluorescence in *Sp5* (RNAi) epidermal (left) or gastrodermal (right) *HySp5*-3169:GFP animals as depicted in (**A**). See Appendix A. (**D**) RNAi procedure applied in experiments depicted in panels D and E. Q-PCR quantification of *Sp5*, *GFP*, *Wnt3* and *β-catenin* transcripts in the apical (100–80%), central body column (BC, 80–30%) and basal (30–0%) regions of epidermal (left) and gastrodermal (right) *HySp5*-3169:GFP animals dissected two days post-EP2 (2d pEP2). ns: non-significant value, other statistical values as indicated in Materials & Methods. (**E**) GFP (green) and mCherry (red) fluorescence in epidermal (left) or gastrodermal (right) *HySp5*-3169:GFP animals 2d pEP2. White bars indicate areas of ectopic GFP fluorescence along the body column, white arrows point to spots of ectopic gastrodermal GFP fluorescence in tentacles. See Appendix A. Scale bars correspond to 250 µm, except in (**B**) where it is 200 µm.

**Figure 6 biomedicines-12-01274-f006:**
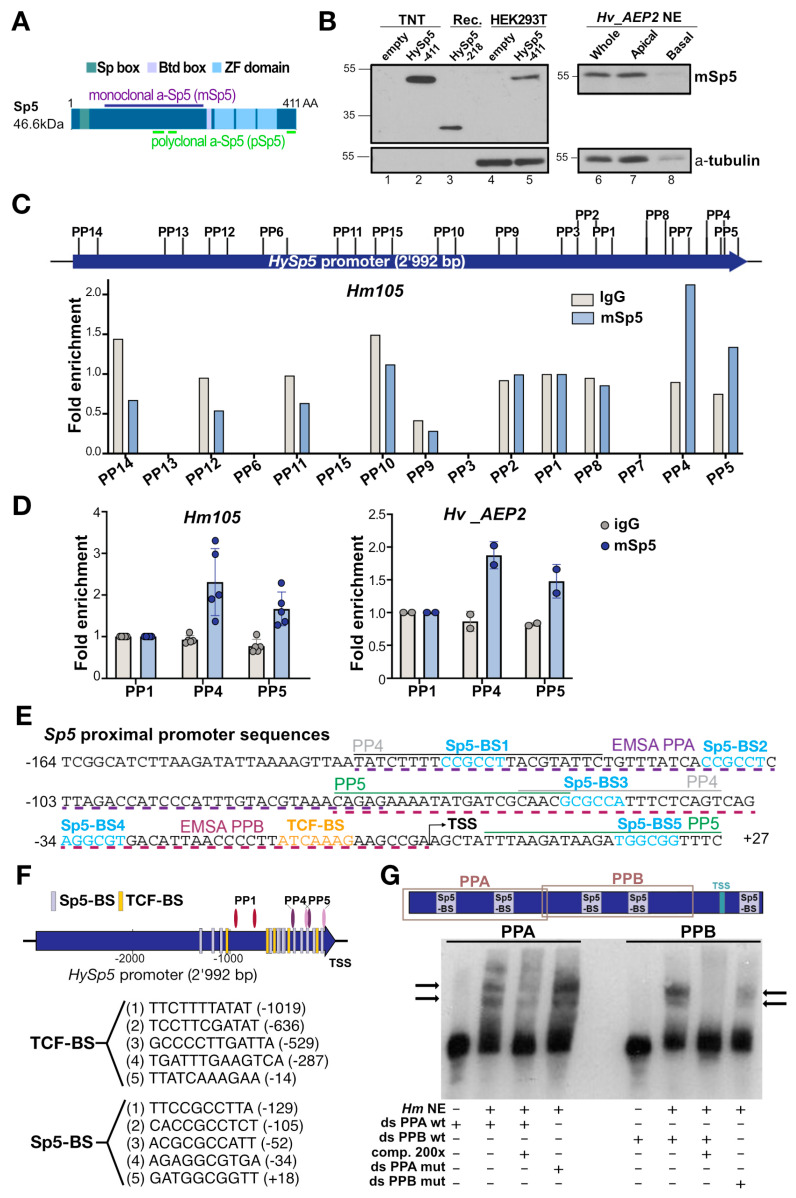
CHIP-qPCR identification of Sp5-binding sites in the *Hydra Sp5* promoter. (**A**) Structure of the *Hydra* Sp5 protein with the conserved Sp box (green), Buttonhead box (Btd, light purple) and zinc-finger (ZF) domain (light blue); the purple line indicates the 218 AA-long region used to raise the monoclonal anti-Sp5 (mSp5) antibody, the green lines indicated the peptides used to raise the polyclonal anti-Sp5 (pSp5) antibody. (**B**) Western blot using the mSp5 antibody to detect the TNT-produced HySp5 protein (lanes 1, 2), the HySp5-218 recombinant protein used to raise mSp5 (lane 3), HySp5 expressed in HEK293T cells (lanes 4, 5) or HySp5 present in *Hv_AEP2* nuclear extracts prepared from whole animals (lane 6), from apical (100–50%, lane 7) and basal (50–0%, lane 8) halves. (**C**) Schematic view of the 15 regions (see 15 pairs of primers in Appendix A) tested along the 2992 bp-long *Sp5* promoter by ChIP-qPCR using *Hm105* extracts and the mSp5 antibody. Significant enrichment is noted only in regions PP4 and PP5. (**D**) Similar ChIP-qPCR enrichment in regions PP4 and PP5 obtained with mSp5 antibody when *Hm105* (left) or *Hv_AEP2* (right) extracts are used. See Appendix A. (**E**) Proximal *HySp5* promoter sequence (−162 to +29), which contains the transcriptional start sites (TSS) identified in *Hv_AEP* (TSS1 +1) and *Hm105* (TSS2 −149, see Appendix A), five Sp5 binding sites (SP5-BS, light blue), a single TCF binding site (TCF-BS, orange), the PP4 (grey) and PP5 (green) primers used for ChIP-qPCR. In the same region, the PPA (−135 to −67) and PPB (−71 to +2) stretches, underlined with purple (PPA) and pink (PPB) dashed lines, respectively, were used in Electro-Mobility Shift Assay (EMSA). (**F**) Schematic map of the *HySp5* 2992 bp promoter region indicating the predicted TSS1, the clustered TCF-BS and Sp5-BS (light purple and orange bars, respectively), the PP1, PP4 and PP5 primer pairs. Sequences of the TCF-BS and Sp5-BS identified in the *HySp5* promoter. (**G**) EMSA showing a shift (arrows) of the PPA and PPB ds-DNAs incubated with *Hm105* NEs. Comp.: unlabeled ds-PPA (left) or ds-PPB (right) added as competitor 200 fold in excess during the incubation; mut: mutated.

**Figure 7 biomedicines-12-01274-f007:**
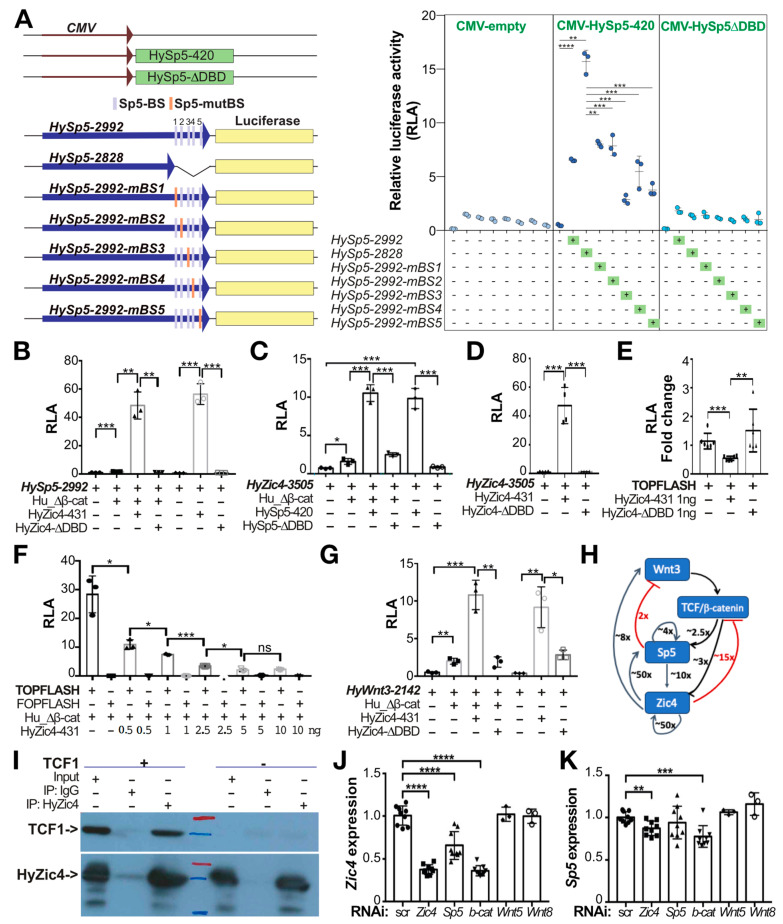
Functional analysis of the *Hydra Sp5*, *Zic4* and *Wnt3* promoters. (**A**–**G**) Luciferase reporter assays performed in HEK293T cells to measure the Relative Luciferase Activity (RLA) driven by various promoters when *Hydra* proteins are co-expressed. Each data point represents one biologically independent experiment. (**A**) RLA levels driven by the *HySp5* promoter either when 2992 bp long (*HySp5-2992*), or when deleted from its proximal region (*HySp5-2828*) that contains five Sp5-binding sites (Sp5BS), or when one of these 5 Sp5BSs is mutated (*HySp5-2992-mBS1*, *HySp5-2992-mBS2*, …). Each construct was tested either in the absence of any protein co-expressed (CMV-empty), or in the presence of co-expressed proteins, full-length Sp5 (HySp5-420) or Sp5 lacking its DNA-Binding Domain (HySp5-ΔDBD). (**B**) RLA levels driven by the *HySp5-2992* promoter when the full-length HyZic4 (HyZic4-431) or the truncated HyZic4 lacking its DNA-Binding Domain (HyZic4-ΔDBD) are co-expressed. (**C**,**D**) RLA levels driven by the *HyZic4-3505* promoter when full-length or truncated HySp5 (HySp5–420, HySp5-ΔDBD in (**C**)) or full-length or truncated HyZic4 (HyZic4-431, HyZic4-ΔDBD in (**D**)) are co-expressed. (**E**,**F**) RLA levels driven by the TOPFlash or FOPFLASH reporter constructs that contain 6× TCF-binding sites either consensus or mutated when HyZic4-431 (**E**,**F**) or HyZic4-ΔDBD (**E**) are co-expressed. (**G**) RLA levels driven by the *Wnt3-2142* promoter when HyZic4-431 or HyZic4-ΔDBD are co-expressed. (**H**) Diagram showing the regulations detected in HEK293T cells on the *Hydra Wnt3*, *Sp5* and *Zic4* upstream sequences when the human β-catenin and/or HySp5 and HyZic4 proteins are co-expressed (this work, [33,34]). (**I**) Immunoprecipitation (IP) of HA-tagged HyZic4-431 expressed in HEK293T cells together or not with huTCF1. IP was performed with an anti-HA antibody and co-IP products were detected with the anti-TCF1 antibody. Same results were obtained in two independent experiments. (**J**,**K**) *Zic4* (**J**) and *Sp5* (**K**) transcript levels measured by qPCR in *Hv_Basel* animals exposed twice to scrambled (scr) RNAs or *Zic4*, *Sp5*, *β-catenin (b-cat)*, *Wnt5* or *Wnt8* siRNAs. Levels are normalized to those measured in control animals exposed to scr RNAs. In all panels, error bars indicate Standard Deviations and statistical *p* values are as indicated in Materials & Methods (unpaired *t*-test).

**Figure 8 biomedicines-12-01274-f008:**
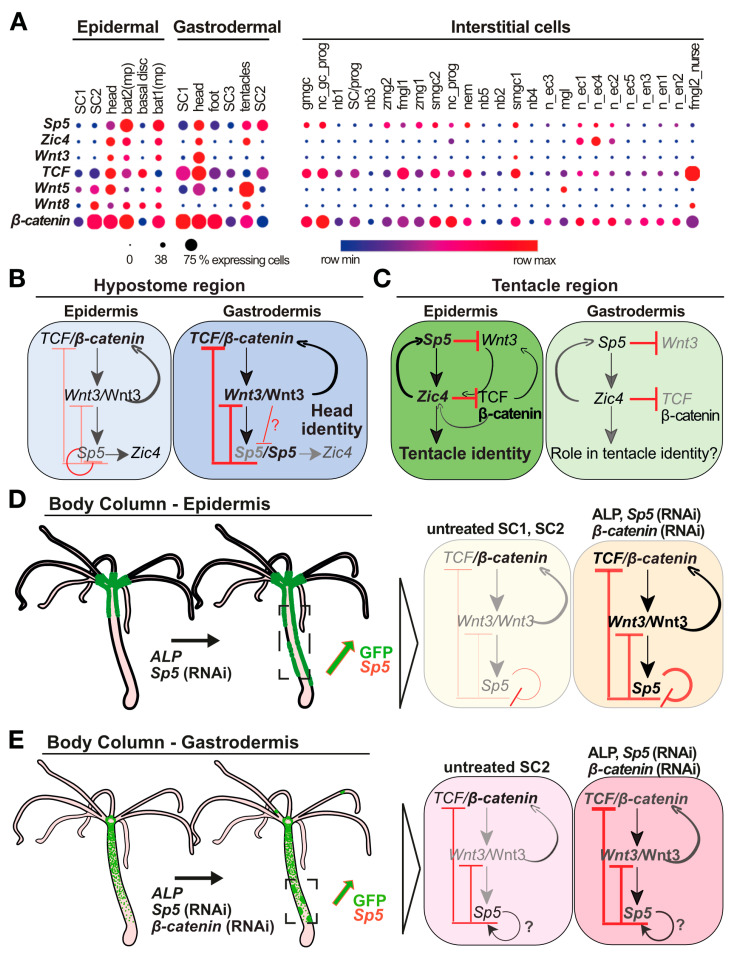
Schematic summary of the layer-specific *Sp5* regulation along the *Hydra* body axis. (**A**) Dot plot view of *Zic4*, *Wnt3*, *Sp5*, *TCF*, *Wnt5A*, *Wnt8* and *β-catenin* expression in cells from the epithelial lineages, either epidermal or gastrodermal, and the interstitial lineage as deduced from *Hydra* single-cell transcriptome analysis [35]. Along the central body column, single-cell sequencing has identified two populations of epithelial stem cells in the epidermis (SC1 and SC2) and three in the gastrodermis (SC1, SC2 and SC3). See other abbreviations in Appendix A. (**B**,**C**) Schematic representation of the predicted genetic regulation network (GRN) at work in the epidermal and gastrodermal layers of the hypostome (**B**) and tentacle (**C**) regions in *Hydra*. (**D**,**E**) Schematic view of GFP fluorescence (green), *Sp5* expression and predicted GRNs at work in the epidermal (**D**) and gastrodermal (**E**) layers of the body column (BC) of *HySp5-3169:GFP* transgenic animals, either maintained in homeostatic conditions (left) or ALP-treated, or knocked-down for *Sp5* or *β-catenin* (right). Bold letters, black arrows and thick red bars indicate a stronger activity.

**Table 1 biomedicines-12-01274-t001:** Zic-binding sites present in chordates’ genes and in the upstream sequences of the *Sp5, Wnt3* and *Zic4 Hydra* genes. *CI: Ciona intestinalis* (ascidian), *HR: Halocynthia roretzi* (ascidian), *HS: Homo sapiens*, *HV: Hydra vulgaris*, *MM: mus musculus*.

Species	Target Genes		Zic TF	ZIC-Binding Site Sequence	References
** *MM* **	*Gli*—consensus		Zic1–5	TGGGTGGTC	[46,47]
** *MM* **	*L* *amin A/C*		Zic1	CCACCCCCT	[48]
** *MM* **	*M* *ath1*		Zic1	GCTCCCCGGGGAGCT	[49]
** *MM* **	*A* *poE*	_A	Zic1/2	GGACTGTGGGGGGTGGTCAA	[50]
		_B		AAACTGTGGGGGGTGGTCAA	
		_C		GGACTGTGGGGGGTGAAAAA	
		_D		CTATCCCTGGGGGAGGGGGC	
** *HS* **	*D1A*		ZIC2	CCCCCAGGGCA	[51]
** *MM* **	*CamK II*		Zic2	GTGTGGGC	[52]
** *MM* **	*Pax3*		Zic2	CTGCTGGGG	[53]
** *MM* **	*Oct4*		Zic2	2550~−2430	[54]
** *HS* **	*a-ACTIN*		ZIC3	GGAGGG	[55]
** *MM* **	*Nanog-C, Nanog-T*		Zic3	CC(C/T)GCTGGG CCTGCTGGG	[56]
** *HS* **	*E-CADHERIN (CDH1)*		ZIC5	−283~−71	[57]
** *CI* **	*B* *rachyury*		ZicL	CCAGCTGTG	[58]
** *CI* **	*E(SPL)/HAIRY-B*		Macho1	5′-GCCCCCCGCTG-3′	[59]
** *HR* **	*synthetic*		Macho1	GACCCCCCA	[60]
** *HV* **	*Sp5*	*−*672	Zic	AACCTGGCCTGC	[33,34]
		*−*383		GGCAGGTGCCGGC	this work (Figure 7 and Appendix A)
** *HV* **	*Wnt3*	−1781	Zic	GCCCGCGCTCTCC	this work (Figure 7 and Appendix A)
		*−*1177		GACAGCGGGTG	
** *HV* **	*Zic4*	−1863	Zic4	ATCGCCCCCTCTCGCT	[34]
		−1825		ACGGGCATTGGCGTGA	this work (Figure 7 and Appendix A)
		−1736		GAGGTGACCCATGCTG	
		−1291		AGGAAAGGGGTGCTACA	
		−107 +77		TGCTCCCGT TACCCCGCTA	

## Data Availability

Data are contained within the article and Appendix A.

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
