# Peer review of "The Wnt/β-catenin/TCF/Sp5/Zic4 Gene Network That Regulates Head Organizer Activity in Hydra Is Differentially Regulated in Epidermis and Gastrodermis"

_biomedicines, 2024, doi:10.3390/biomedicines12061274_

Round 1

Reviewer 1 Report

Comments and Suggestions for Authors

The paper by Ollé et al. is interesting, original and addresses an under-explored question. The authors experimentally investigate not just the differential expression of a gene in different tissues, but the different architecture of genetic regulatory networks in two germ layers, ectoderm and endoderm. The authors have made extensive research using a variety of techniques. However, in order to be published, the article needs to be improved and the presentation of some of the results obtained by the authors needs to be modified.

The entire work is based on the authors' transgenic Hydra strains (clonally propagated F0 generation polyps) expressing the Act-1388:mCherry_HySp5-3169:eGFP construct in either ectoderm or endoderm. Fig. 1 shows images of polyps from these strains - in vivo imaging (Fig. 1E, F) and the results of labelling the GFP protein with antibodies (Fig. 1G, H).

First, there is no direct evidence that the endogenous genes and the transgenes are expressed in the same cells. I would recommend using a combination of fluorescent in situ hybridisation (to label endogenous HySp5) and immunofluorescence (to label GFP protein) to confirm that transgene expression recapitulates endogenous HySp5 expression.

I am also concerned about the visualization of the data. Unfortunately, the ectoderm, endoderm and the boundary between them are not visible in the images (Figure 1E, F, G, H). It is difficult to improve the quality of in vivo imaging. However, the quality of images obtained by immunolabelling and confocal microscopy should be improved. A longitudinal optical (or histological) section along the midline of the polyp body is required to verify that GFP is localized in only one of the body layers or that its pattern differs between body layers. This should clearly show the ectoderm and endoderm separated by the mesoglea, as well as the gastric cavity. To improve the quality of visualisation, it is useful to label nuclei and F-actin together with GFP and mCherry.

The same is true for the quality of the images illustrating the results of insitu hybridisation. As far as I know, the insitu hybridisation protocol for Hydra is very well developed. Unfortunately, of the images presented in the ms, only Fig. 2A, B clearly show both the specific expression patterns of the genes studied and the localization of their expression in the ectoderm or endoderm. A special problem is the visualization of the double insitu hybridisation in Fig. 3B. The pattern of the FLUO-labelled riboprobe detected with Fast Red is blurred in many images. The schematics presented show more detail than is visible in the original images. I have no doubt that all this detail is there, but the quality of the images should be improved. If ectodermal or endodermal expression is demonstrated, it is recommended that the mesoglea be in focus. However, cryosections or histological sections would be the optimal choice.

The text of the manuscript should also be revised. The abstract is somewhat puzzling, as the authors have attempted to include all the results obtained and methods used. There is a term that is not generally accepted ("pseudobud"), and "human HEK293T cells" appears without explanation, although the model object of the ms is Hydra.

[lines 20-22: "Sp5(RNAi) highlights the negative autoregulation of Sp5 in epidermis, involving direct binding of Sp5 to its own promoter as observed in human HEK293T cells. In these cells, HyZic4, which can interact with huTCF1, regulates Wnt3 negatively and Sp5 positively."]

It is therefore necessary to undertake a thorough revision of the abstract, ensuring that it presents only the key findings and conclusions. It is also important to ensure that the abstract can be understood without reading the entire article.

A few minor corrections should be made to the introduction.

Lines 33 - 35: "Its anatomy is simple, basically a gastric tube composed of two myoepithelial layers known as the epidermis and gastrodermis along a single oral-aboral axis (Figure 1A)."

- Figure 1A does not show the epidermis and gastrodermis.

Lines 39-41: "Indeed, she showed that tissues isolated from the head of intact animals, from the head regenerating tip or from the presumptive head of the developing bud, can instruct and recruit cells from the body column of the host to induce the formation of an ectopic head, a property later named organiser activity."

- Ethel Browne did not introduce the term "organizer". Authors should refer here to Spemann, Mangold, 1924.

Lines 49-51: "The principle of organizer activity was later shown to be also at work during embryonic development in vertebrates, initially in gastrulae [9,10] and later on during appendage and hindbrain development [11-14]."

- It is important to distinguish between axial organizers and inducers (not every inducer is an organizer).

Lines 88 - 99: There should be a reference to Figure 1A. It would be useful to include a schematic representation of the expression patterns of the genes that are discussed in the text - Wnt3, Sp5 and Zic4.

In general, the text should be well proofread, as there are occasionally missing words.

Author Response

Reply to comments of Reviewer 1

The paper by Ollé et al. is interesting, original and addresses an under-explored question. The authors experimentally investigate not just the differential expression of a gene in different tissues, but the different architecture of genetic regulatory networks in two germ layers, ectoderm and endoderm. The authors have made extensive research using a variety of techniques. However, in order to be published, the article needs to be improved and the presentation of some of the results obtained by the authors needs to be modified.

BG et al.: We would like to thank Reviewer 1 for their careful reading of our article and their constructive comments. Below we respond to each of their criticisms, and have taken on board all their suggestions.

The entire work is based on the authors' transgenic Hydra strains (clonally propagated F0 generation polyps) expressing the Act-1388:mCherry_HySp5-3169:eGFP construct in either ectoderm or endoderm. Fig. 1 shows images of polyps from these strains - in vivo imaging (Fig. 1E, F) and the results of labelling the GFP protein with antibodies (Fig. 1G, H).

First, there is no direct evidence that the endogenous genes and the transgenes are expressed in the same cells. I would recommend using a combination of fluorescent in situ hybridisation (to label endogenous HySp5) and immunofluorescence (to label GFP protein) to confirm that transgene expression recapitulates endogenous HySp5 expression.

BG et al.: At an early stage of this study, we have made this experiment of detecting in the same animals the endogenous HySp5 expression and the GFP protein. At that time, we used a chromogenic detection of Sp5 expression and immunofluorescence to detect GFP. Epithelial cells, either from the epidermis or from the gastrodermis, indeed co-express Sp5 and GFP and we provide in the revised version of this article a new supplementary figure (Figure S3) showing the imaging of animals with co-detection of bright field Sp5 expression and fluorescent GFP (panels C and D). The quality of fluorescence imaging is not of the best as confocal imaging is not possible for such codetection. It would be indeed necessary to repeat this experiment using, as suggested by Reviewer 1, fluorescent in situ and fluorescent immunodetection. Given the time constraint, this is unfortunately not feasible, but the new images added in Figure S3 now provide good evidence of colocalisation.

I am also concerned about the visualization of the data. Unfortunately, the ectoderm, endoderm and the boundary between them are not visible in the images (Figure 1E, F, G, H). It is difficult to improve the quality of in vivo imaging. However, the quality of images obtained by immunolabelling and confocal microscopy should be improved. A longitudinal optical (or histological) section along the midline of the polyp body is required to verify that GFP is localized in only one of the body layers or that its pattern differs between body layers. This should clearly show the ectoderm and endoderm separated by the mesoglea, as well as the gastric cavity. To improve the quality of visualisation, it is useful to label nuclei and F-actin together with GFP and mCherry.

BG et al.: We agree with Reviewer 1 that live imaging of the animals is difficult as we can only keep them immobile for very short periods of time. We have now added to Figure 1 (panels 1C, 1D) as well as in Supplemental Figures (Figure S3, panels A and B) new images of live animals with optical sections that show the clear boundary between the two layers. As expected, GFP fluorescence is present exclusively in epithelial cells of the epidermis in epidermal HySp5-3169:GFP transgenic animals, whereas in gastrodermal HySp5-3169:GFP transgenic animals, GFP fluorescence is present exclusively in gastrodermal epithelial cells. These images prove that each transgenic line is indeed specific to one epithelial layer, the epidermis or gastrodermis.

The same is true for the quality of the images illustrating the results of insitu hybridisation. As far as I know, the insitu hybridisation protocol for Hydra is very well developed. Unfortunately, of the images presented in the ms, only Fig. 2A, B clearly show both the specific expression patterns of the genes studied and the localization of their expression in the ectoderm or endoderm. A special problem is the visualization of the double insitu hybridisation in Fig. 3B. The pattern of the FLUO-labelled riboprobe detected with Fast Red is blurred in many images. The schematics presented show more detail than is visible in the original images. I have no doubt that all this detail is there, but the quality of the images should be improved. If ectodermal or endodermal expression is demonstrated, it is recommended that the mesoglea be in focus. However, cryosections or histological sections would be the optimal choice.

BG et al.: We agree with Reviewer 1 that the double in situ shown in Figure 3 with colorimetric detection using a combination of NBT/BCIP and Fast Red staining shows a rather diffuse pattern. But this pattern reflects the fact that after alsterpaullone treatment, many epithelial cells along the body column express GFP, Sp5 or Wnt3, often at different levels explaining for example that above a moderate level of expression of mots epithelial cells that give the blurry aspect, some cells expressing Sp5 or GFP at higher level form circular figures in the epidermis.

For each condition, we present 10 animals (one in the main figure and the complete collection in the supplementary file) from the non-transgenic strain Hv-AEP2 (Figure S7), the transgenic strains HySp5-3169:GFP epidermal and gastrodermal (Figure S8), the transgenic strains HyWnt3-2149:GFP epidermal and gastrodermal (Figure S9). The expression patterns of each animal were analyzed, and to highlight the main features found in most animals of a given condition, we have added diagrams representing these features to help the reader visualize the results. Furthermore, results obtained with HyWnt3-2149:GFP transgenic strains are consistent with those obtained with HySp5-3169:GFP transgenic strains. For all these reasons, we consider these results to be robust.

The text of the manuscript should also be revised. The abstract is somewhat puzzling, as the authors have attempted to include all the results obtained and methods used. There is a term that is not generally accepted ("pseudobud"), and "human HEK293T cells" appears without explanation, although the model object of the ms is Hydra.

BG et al.: We thank Reviewer 1 for these suggestions. Firstly, we have reshaped the abstract so that it contains less detail that is not necessary to understand the results and conclusions of this study. Secondly, we have changed the term “pseudo-buds” to “bud-like structures”, while explaining that these bud-like structures are not identical to “natural buds”, a phenotype never before named in the scientific literature.

[lines 20-22: "Sp5(RNAi) highlights the negative autoregulation of Sp5 in epidermis, involving direct binding of Sp5 to its own promoter as observed in human HEK293T cells. In these cells, HyZic4, which can interact with huTCF1, regulates Wnt3 negatively and Sp5 positively."]

BG et al.: This sentence is now shortened and transform to: “Sp5(RNAi) reveals a negative Sp5 autoregulation in the epidermis but not in the gastrodermis.”

It is therefore necessary to undertake a thorough revision of the abstract, ensuring that it presents only the key findings and conclusions. It is also important to ensure that the abstract can be understood without reading the entire article.

BG et al.: The abstract has been thoroughly revised and can now be understood without reading the entire article.

A few minor corrections should be made to the introduction.

Lines 33 - 35: "Its anatomy is simple, basically a gastric tube composed of two myoepithelial layers known as the epidermis and gastrodermis along a single oral-aboral axis (Figure 1A)."

- Figure 1A does not show the epidermis and gastrodermis.

BG et al.: We have suppressed the reference to Figure 1A in this sentence.

Lines 39-41: "Indeed, she showed that tissues isolated from the head of intact animals, from the head regenerating tip or from the presumptive head of the developing bud, can instruct and recruit cells from the body column of the host to induce the formation of an ectopic head, a property later named organiser activity."

- Ethel Browne did not introduce the term "organizer". Authors should refer here to Spemann, Mangold, 1924.

BG et al.: We have now added the reference to Spemann & Mangold (1924) in this sentence.

Lines 49-51: "The principle of organizer activity was later shown to be also at work during embryonic development in vertebrates, initially in gastrulae [9,10] and later on during appendage and hindbrain development [11-14]."

- It is important to distinguish between axial organizers and inducers (not every inducer is an organizer).

BG et al.: We understand the distinction made by Reviewer 1 between axial organizers and inducers but we do not see organizers active in limb development for example or in appendage regeneration as conceptually different from axial organizers. This debate would need to bring specific arguments that go beyond the scope of this study.

Lines 88 - 99: There should be a reference to Figure 1A. It would be useful to include a schematic representation of the expression patterns of the genes that are discussed in the text - Wnt3, Sp5 and Zic4.

BG et al.: We have modified Figure 1A to make clear the different domains of Wnt3, Sp5 and Zic4 expression and we now refer to this revised scheme in this part of the introduction.

In general, the text should be well proofread, as there are occasionally missing words.

BG et al.: We have carefully re-read the text and have corrected sentences that could be considered as unclear or missing some words.

Reviewer 2 Report

Comments and Suggestions for Authors

This manuscript builds on decades of investigations into the “head organizer” in hydra and provides an important and exhaustive study elucidating the similarity and differences of gene expression in the two epithelial layers.  Multiple techniques were expertly employed to develop a rich data set that is clearly presented.  This work will greatly contribute to the understanding of the workings of the hydra head organizer.  The discussion nicely summarizes these contributions as well as the remaining challenges.  Central to these challenges are the variable developmental responses of different laboratory strains of Hydra vulgaris to the same perturbations (nicely summarized in section 4.6).  This variability has been more-or-less taken for granted by the hydra community for some time, but it raises still-unanswered questions, e.g., how much of this sort of variation is found in natural populations?  Qualifiers like “stronger activity” or “greater penetrance” do not provide much illumination.

The comments that follow are meant to suggest small wording changes:

Lines 31-33: “Hydra is a freshwater hydrozoan polyp known for its exceptional regenerative capacities, able to regrow any missing part of its body, such as a new fully functional head in three to four days after mid-gastric bisection (reviewed in [1]).”  Suggest something like: “…regenerative capacities, including its ability to regrow any missing part of its…”

Introduction: a nice historical summary!

Lines 199-120: “Hydra vulgaris (Hv) from the Basel (Hv_Basel), magnipapillata (Hm-105) or AEP2 (Hv_AEP2) strains [36] were cultured….”  Suggest something like: “Strains of Hydra vulgaris (Hv) from the Basel (Hv_Basel), magnipapillata (Hm-105) or AEP2 119 (Hv_AEP2) [36] were cultured….”

Line 122: “nauplii”: nauplii

Lines 550-551: “In summary, the analysis of epidermal and gastrodermal GFP expression in these four transgenic lines help identify…”: “…the analyses…help…” or “…the analysis…helps…”

Lines 568-569: “This observation suggests that the EP procedure leads to an unspecific stress-induced response that either activates b-catenin regulatory sequences, and/or stabilizes b-catenin transcripts (Figure 4B, Figure S9).”  A noteworthy general observation.

Lines 834-835: “These results show that the HySp5 promoter is submitted to a complex regulation,…”  Suggest: “These results show that the 834 HySp5 promoter is subject to a complex regulation,…” 

Author Response

Reply to comments of Reviewer 2

This manuscript builds on decades of investigations into the “head organizer” in hydra and provides an important and exhaustive study elucidating the similarity and differences of gene expression in the two epithelial layers. Multiple techniques were expertly employed to develop a rich data set that is clearly presented. This work will greatly contribute to the understanding of the workings of the hydra head organizer. The discussion nicely summarizes these contributions as well as the remaining challenges. Central to these challenges are the variable developmental responses of different laboratory strains of Hydra vulgaris to the same perturbations (nicely summarized in section 4.6). This variability has been more-or-less taken for granted by the hydra community for some time, but it raises still-unanswered questions, e.g., how much of this sort of variation is found in natural populations? Qualifiers like “stronger activity” or “greater penetrance” do not provide much illumination.

BG et al.: We would like to thank Reviewer 2 for their careful reading of our article and their positive comments. We have taken into account all their suggestions concerning the wording of the text.

The comments that follow are meant to suggest small wording changes:

Lines 31-33: “Hydra is a freshwater hydrozoan polyp known for its exceptional regenerative capacities, able to regrow any missing part of its body, such as a new fully functional head in three to four days after mid-gastric bisection (reviewed in [1]).” Suggest something like: “...regenerative capacities, including its ability to regrow any missing part of its...”

BG et al.: We have modified the sentence accordingly.

Introduction: a nice historical summary!

Lines 199-120: “Hydra vulgaris (Hv) from the Basel (Hv_Basel), magnipapillata (Hm-105) or AEP2 (Hv_AEP2) strains [36] were cultured....” Suggest something like: “Strains of Hydra vulgaris (Hv) from the Basel (Hv_Basel), magnipapillata (Hm-105) or AEP2 119 (Hv_AEP2) [36] were cultured....”

BG et al.: We have modified the sentence as suggested.

Line 122: “nauplii”: nauplii

BG et al.: “nauplii” is no longer written italics.

Lines 550-551: “In summary, the analysis of epidermal and gastrodermal GFP expression in these four transgenic lines help identify...”: “...the analyses...help...” or “...the analysis...helps...”

BG et al.: we have now modified this sentence to: “In summary, analysis of GFP expression in these four transgenic lines helps identify the layer-specific regulation of Sp5 and Wnt3.”

Lines 568-569: “This observation suggests that the EP procedure leads to an unspecific stress-induced response that either activates b-catenin regulatory sequences, and/or stabilizes b-catenin transcripts (Figure 4B, Figure S9).” A noteworthy general observation.

Lines 834-835: “These results show that the HySp5 promoter is submitted to a complex regulation,...” Suggest: “These results show that the 834 HySp5 promoter is subject to a complex regulation,...”

BG et al.: We have modified the sentence as suggested.
